# Spatial variations of CO₂ fluxes in the Saguenay Fjord (Québec, Canada) and results of a water mixing model

Louise Delaigue[1], Helmuth Thomas[2,3] and Alfonso Mucci[1]

[1]GEOTOP and Department of Earth and Planetary Sciences, McGill University, 3450 University Street, Montreal, QC H3A 0E8, Canada

[2]Department of Oceanography, Dalhousie University, Halifax, Nova Scotia, Canada

[3]Center for Materials and Coastal Research, Helmholtz-Zentrum Geesthacht, Germany

*Correspondence to*: Louise Delaigue (louise.delaigue@mail.mcgill.ca)

**Abstract.** The Saguenay Fjord is a major tributary of the St. Lawrence Estuary and is strongly stratified. A 6-8 m wedge of brackish water typically overlies up to 270 m of seawater. Relative to the St. Lawrence River, the surface waters of the Saguenay Fjord are less alkaline and host higher dissolved organic carbon (DOC) concentrations. In view of the latter, surface waters of the fjord are expected to be a net source of $CO_2$ to the atmosphere, as they partly originate from the flushing of organic-rich soil porewaters. Nonetheless, the $CO_2$ dynamics in the fjord are modulated with the rising tide by the intrusion, at the surface, of brackish water from the Upper St. Lawrence Estuary, as well as an overflow of mixed seawater over the shallow sill from the Lower St. Lawrence Estuary. Using geochemical and isotopic tracers, in combination with an optimization multi-parameter algorithm (OMP), we determined the relative contribution of known source-waters to the water column in the Saguenay Fjord, including waters that originate from the Lower St. Lawrence Estuary and replenish the fjord's deep basins. These results, when combined to a conservative mixing model and compared to field measurements, serve to identify the dominant factors, other than physical mixing, such as biological activity (photosynthesis, respiration) and gas exchange at the air-water interface, that impact the water properties (e.g., pH, $pCO_2$) of the fjord. Results indicate that the fjord's surface waters are a net source of $CO_2$ to the atmosphere during periods of high freshwater discharge (e.g., spring freshet) whereas they serve as a net sink of atmospheric $CO_2$ when their practical salinity exceeds $\sim$ 5-10.

## 1 Introduction

Anthropogenic emissions of carbon dioxide ($CO_2$) have recently propelled atmospheric $CO_2$ concentrations above the 410 ppm mark, the highest concentration recorded in the past 3 million years (Willeit et al., 2019). The oceans, the largest $CO_2$ reservoir on Earth, have taken up ca. 30% of the anthropogenic $CO_2$ emitted to the atmosphere since the beginning of the industrial era (Feely et al., 2004; Brewer and Peltzer, 2009; Doney et al., 2009; Orr, 2011, Friedlingstein et al., 2019), mitigating the impact of this greenhouse gas on global warming (Sabine et al., 2004). On the other hand, the uptake of $CO_2$ by the oceans has led to modifications of the seawater carbonate chemistry and a decline in the average surface ocean pH by ~0.1 units since pre-industrial times, a phenomenon dubbed ocean acidification (Caldeira and Wickett, 2005). According to the Intergovernmental Panel on Climate Change (IPCC) "business as usual" emissions scenario IS92a and general circulation models, atmospheric $CO_2$ levels may reach 800 ppm by 2100, lowering the pH of the surface oceans by an additional 0.3-0.4 units, at a rate that is unprecedented in

the geological record (Caldeira and Wickett, 2005; Hönisch et al., 2012; Rhein et al., 2013). The growing concern about the impacts of anthropogenic $CO_2$ emissions on climate as well as marine and terrestrial ecosystems calls for a meticulous quantification of organic and inorganic carbon fluxes, especially in coastal environments, including fjords, a major but poorly quantified component of the global carbon cycle and budget (Bauer et al., 2013; Najjar et al., 2018). Meaningful predictions of the effects of climate change on future fluxes are intricate given the very large uncertainty associated with present-day air-sea $CO_2$ flux estimates in coastal waters, including rivers, estuaries, tidal wetlands, and the continental shelf (Bauer et al., 2013; Najjar et al., 2018). The coastal ocean occupies only ~7% of the global ocean surface area, but plays a major role in biogeochemical cycles because it (1) receives massive inputs of terrestrial organic matter and nutrients through continental runoff and groundwater discharge; (2) exchanges matter and energy with the open ocean; and (3) is one of the most geochemically and biologically active areas of the biosphere, accounting for significant fractions of marine primary production (~14 to 30%), organic matter burial (~80%), sedimentary mineralization (~90%), and calcium carbonate deposition (~50%) (Gattuso et al., 1998).

Although the carbon cycle of the coastal ocean is acknowledged to be a major component of the global carbon cycle and budget, accurate quantification of organic and inorganic carbon cycling and fluxes in the coastal ocean — where land, ocean and atmosphere interact — remains challenging (Bauer et al., 2013; Najjar et al., 2018). Constraining the exchanges and fates of different forms of carbon along the land—ocean continuum is so far incomplete, owing to limited data coverage and large physical and biogeochemical variability within and between coastal subsystems (e.g., hydrological and geomorphological differences, differences in the magnitude and stoichiometry of organic matter inputs). Hence, owing to limited data coverage and suspicious upscaling due to the large physical and biogeochemical variability within and between coastal subsystems, there remains a debate as to whether coastal waters are net sources or sinks of atmospheric $CO_2$. Recent compilations of worldwide $CO_2$ partial pressure (p$CO_2$) measurements indicate that most open shelves in temperate and high-latitudes are sinks of atmospheric $CO_2$ whereas low-latitude shelves and most estuaries are sources (Chen and Borges, 2009; Cai, 2011; Chen et al., 2013). As noted by Bauer et al. (2013), estuaries are transitional aquatic environments that can be riverine or marine dominated and, thus, they typically display strong gradients in biogeochemical properties and processes as they flow seaward. Chen et al. (2013) reported that the strength of estuarine sources typically decreases with increasing salinity. However, marsh-dominated estuaries, in which active microbial decomposition of organic matter occurs in the intertidal zone, are strong sources of $CO_2$ (Cai, 2011).

High latitude waters such as the Arctic Ocean have recently been the foci of much research, while coastal, seasonally ice-covered aquatic environments, such as the Saguenay Fjord, that display comparable inter-annual and climatic sea-ice cover variabilities but are much more accessible, have been neglected (Bourgault et al., 2012). Characteristics of Arctic coastal ecosystems are found in the Saguenay Fjord, including the presence of many species of plankton, fish, birds and marine mammals as well as important freshwater inputs and the presence of seasonal ice cover (Bourgault et al., 2012). Fjords stand amongst the most productive ecosystems on the planet, while they have a yet unexplored role in regional and global carbon cycles as part of the estuarine family (Juul-Pedersen et al., 2015). They are crucial hotspots for organic carbon (mostly terrestrial) burial and account for nearly 11% of the annual organic carbon burial flux in marine sediments, while covering only 0.12% of oceans' surface

(Rysgaard et al., 2012; Smith et al., 2015). In other words, organic carbon burial rates in fjords are a hundred times faster than the average rate in the global ocean. Rates of organic carbon burial provide insights on the mechanism that controls atmospheric $O_2$ and $CO_2$ concentrations over geological timescales (Smith et al., 2015).

This study presents 1) the relative contribution of known source waters to the water column in the fjord, estimated from the solution of an optimization multi-parameter algorithm (OMP) using geochemical and isotopic tracers, and 2) results of a conservative mixing model, based on results of the OMP analysis and from which theoretical surface-water $pCO_2$ values are derived and then compared to field measurements. The latter comparison serves to identify the dominant factors, other than physical mixing (i.e., biological activity, gas exchange), that impact the $CO_2$

fluxes at the air-sea interface and modulate their direction and intensity throughout the fjord (i.e. whether it is a source or a sink of $CO_2$ to the atmosphere).

## 2 Data and methods

### 2.1 Study site characteristics

Located in the subarctic region of Québec, eastern Canada, the Saguenay Fjord is up to 275 m deep, 110 km

long and has an average width of 2 km, with a 1.1 km wide mouth where it connects to the head of the Lower St. Lawrence Estuary (Fig. 1.a). The fjord's bathymetry includes three basins bound by three sills (Fig. 1.b). The first one, at a depth of ~20 m, is located at its mouth near Tadoussac and controls the overall dynamics of the fjord. The second is located 18 km further upstream and sits at a depth of 60 m, while the third one is found another 32 km further upstream and rises to a depth of 115 m. The fjord's drainage basin is 78,000 km$^2$ and is part of the greater St. Lawrence

drainage basin (Smith and Walton, 1980), forming a hydrographic system, along with the Great Lakes, of more than 1.36 million km$^2$.

Tributaries to the Saguenay Fjord include the Saguenay, Éternité and Sainte-Marguerite Rivers (Fig. 1.a). The Saguenay River is the main outlet from the Saint-Jean Lake, and flows into the North Arm of the fjord near St. Fulgence (Fig. 1.a) with a mean freshwater discharge of ~1200 m$^3$ s$^{-1}$ (Bélanger, 2003). Two other local, minor

tributaries, the Rivière-à-Mars (95 km long, mean discharge ~8 m$^3$ s$^{-1}$) and the Rivière des Ha! Ha! (35 km long, mean discharge ~15 m$^3$ s$^{-1}$) discharge into the Baie des Ha! Ha!, a distinct feature of the Saguenay Fjord (Fig. 1.a). Finally, the fjord receives denser marine waters from the St. Lawrence Estuary, filling the bottom of the three basins, as these waters episodically overflow the entrance sill (Therriault and Lacroix, 1975; Stacey and Gratton, 2001; Bélanger, 2003; Belzile et al., 2016). According to Seibert et al. (1979), the tidal amplitude at the mouth of the fjord near

Tadoussac averages 4.0 m and increases slightly toward the head of the fjord (4.3 m near Port Alfred). Spring tides may reach an amplitude of 6 m.

The overflow and the intrusion of marine waters from the St. Lawrence Estuary generate a sharp halocline, leading to a simplified two-layer stratification in the fjord (Fig 1.b). The tidally-modulated intrusion of marine waters from the St. Lawrence Estuary into the Saguenay Fjord, as well as the outflow of the fjord into the estuary, have a

major influence on the water column stratification and circulation in the Saguenay Fjord and at its mouth (Belzile et al., 2016; Mucci et al., 2017). In other words, the properties of the uppermost 100 m of the water column in the

adjacent estuary are critical in determining the water stratification in the Saguenay Fjord, since salinity and temperature control the density of waters that spill over the sill and fill the fjord's deep basins (Belzile et al., 2016). During most of the ice-free season, the St. Lawrence Estuary is characterized by three distinct layers: (1) a relatively warm and salty bottom layer (LSLE, 4°C < T < 6°C, 34 < $S_P$ < 34.6; where T stands for temperature and $S_P$ refers to practical salinity) that originates from mixing, on the continental shelf, of northwestern Atlantic and Labrador Current waters, (2) a cold intermediate layer (CIL, 30 - 150 m deep; -1°C < T < 2°C, 31.5 < $S_P$ < 33) that forms in the Gulf of St. Lawrence in the winter and flows landward, and (3) a warm brackish surface layer (0 - 30 m deep, -0.6°C < T < 12°C, 25 < $S_P$ < 32) that results from the mixture of freshwater from various tributaries (mostly the St. Lawrence and Saguenay Rivers, but also north shore rivers such as the Betsiamites, Romaine and Manicouagan) and seawater and flows seaward to ultimately form the Gaspé Current (Dickie and Trites, 1983; El-Sabh and Silverberg, 1990; Gilbert and Pettigrew, 1997). Seasonal variations greatly affect the properties of the surface layer which merges with the intermediate layer during winter, as temperature and salinity change with atmospheric and buoyancy forcing and the contribution from tributaries decreases during winter months (Galbraith, 2006).

Likewise, the Saguenay Fjord is characterized by a strongly stratified water column that includes at least two water masses: (1) a warm, shallow layer, the Saguenay Shallow Water (SSW; 0°C < T < 16.8°C, 0.2 < $S_P$ < 26.9), that lies above (2) the Saguenay Deep Water (SDW; 0.9°C < T < 4.0°C, 27.3 < $S_P$ < 29.8). The SDW most likely forms from a mixture of surface fjord water, St. Lawrence River waters and the St. Lawrence Estuary Cold Intermediate Layer (CIL), when the latter spills over the entrance sill at the mouth of the fjord (Bourgault et al., 2012; Belzile et al., 2016). Nonetheless, our study shows that, because the Saguenay Fjord is a relatively deep fjord with multiple sills, the vertical structure of the water column is far more complex than described above.

**2.2 Water-column sampling**

The data presented in this paper were gathered on five cruises, between the years 2014 and 2018 aboard the R/V Coriolis II, in late spring (May 2016 and May 2018) and early summer (June 2017), as well as early and late fall (September 2014 and November 2017). Sampling of the water column was carried out with a rosette system along the central axis of the Saguenay Fjord, between St. Fulgence and the mouth of the fjord, including the Baie des Ha! Ha!. Stations in the St. Lawrence Estuary, near the mouth of the fjord, were also sampled. The sampling locations are identified in Fig. 1.a. The surface water of the Saguenay River was sampled, with a rope and bucket in 2013 and 2017, from the Dubuc Bridge that joins Chicoutimi and Chicoutimi-Nord, to determine the chemical characteristics of the freshwater Saguenay River end-member.

The rosette system (12 x 12-L Niskin bottles) was equipped with a Seabird 911Plus conductivity-temperature-depth (CTD) probe, a Seabird® SEB-43 oxygen probe, a WETLabs® C-Star transmissometer and a Seapoint® fluorometer. The Niskin bottles were closed at discrete depths as the rosette was raised from the bottom, typically at the surface (2-3 m), 25 m, 50 m, 75 m, 100 m, and at 50 m intervals to the bottom (or within 10 m of the bottom). Samples were taken directly from the bottles for dissolved oxygen (DO), $pH_{NBS}$ and/or $pH_T$, total alkalinity (TA), dissolved inorganic carbon (DIC), dissolved silicate (DSi), practical salinity ($S_P$), and the stable oxygen isotopic composition of the water ($\delta^{18}O_{water}$). Water samples destined for pH measurements were transferred to 125 mL plastic

bottles without headspace whereas TA and TA/DIC samples were stored in, respectively, 250 mL and 500 mL glass bottles. TA and TA/DIC samples were poisoned with a few crystals of mercuric chloride ($HgCl_2$) and bottles were sealed using a ground-glass stopper and Apiezon® Type-M high-vacuum grease. $\delta^{18}O_{water}$ and $S_P$ samples were stored in 13 mL plastic screw-cap test tubes.

Direct measurements of surface water (~ 2 m) $pCO_2$ were carried out using a $CO_2$-Pro CV (Pro-Oceanus, Bridgewater, NS) probe in May 2018. The $CO_2$-Pro CV probe operates through rapid diffusion of gases through a supported semi-permeable membrane to a thermostated cell in which the $CO_2$ mole fraction is quantified by a non-dispersive infrared detector (NDIR) that was factory calibrated using standard trace gas mixtures. The instrument was operated in continuous mode, with measurements taken nearly every 7 seconds. Stable $pCO_2$ values were achieved after a 15-minute equilibration period and averaged over the next 20 minutes. Relative standard deviations over this period were typically on the order of 0.2 to 6% but were on the order of 0.1% in a stable water mass at 220 m depth, implying that deviations recorded at the surface likely reflected natural variations over the period of sampling as the ship drifted with the current. The manufacturer claims a 1% accuracy, but the performance of the instrument may be even better (Hunt et al., 2017).

Total freshwater discharge data of the Saguenay River were provided by Rio Tinto Alcan (a multinational aluminium smelter/producer that manages its own hydroelectric dam on the Saguenay River) from their bank stabilization programme. Data for the relevant sampling days in September 2014, May 2016, June 2017, November 2017 and May 2018 were taken from the Shipshaw and Chute-à-Caron monitoring stations.

**2.3 Analytical procedures**

T and $S_P$ were determined in-situ using the CTD probe. The conductivity probe was calibrated by the manufacturer over the winter prior to the cruises. In addition, the $S_P$ of surface waters was determined by potentiometric argentometric titration at McGill University and calibration of the $AgNO_3$ titrant with IAPSO standard seawater. The reproducibility of these measurements is typically better than ± 0.5%.

$pH_T$ was determined spectrophotometrically on board, on the total hydrogen ion concentration scale for saline waters ($S_P > 5$), using phenol red and purified m-cresol purple as indicators and a Hewlett-Packard UV-visible diode array spectrophotometer (HP-8453A) with a 5-cm quartz cell, after thermal equilibration of the sample in a constant temperature bath at 25°C ± 0.1. The salinity-dependence of the dissociation constants and molar absorptivities of the indicators were taken from Robert-Baldo et al. (1985) for phenol red and from Clayton and Byrne (1993) for m-cresol purple. The salinity-dependence of the phenol red indicator dissociation constant and molar absorptivities was extended (from $S_P = 5$ to 35; Bellis, 2002) to encompass the range of salinities encountered in this study, but computed $pH_T$ values from the revised fit were not significantly different from those obtained with the relationship provided by Robert-Baldo et al. (1985). Results computed from these parameters yielded results that were more similar to each other as well as to potentiometric glass electrode measurements than the revised equation for the purified m-cresol purple provided by Douglas and Byrne (2017). The pH of low-salinity waters ($S_P < 5$) was determined potentiometrically on board at 25°C, on the NIST (formerly NBS) scale ($pH_{NBS}$), using a Radiometer Analytical® (GK2401C) combination glass electrode connected to a Radiometer Analytical® pH/millivoltmeter (PHM84). A

calibration of the electrode was completed prior to and after each measurement, using three NIST-traceable buffer solutions: pH-4.00, pH-7.00 and pH-10.00, at 25°C. The Nernstian slope was then obtained from the least-squares fit of the electrode response to the NIST buffer values. For waters with $S_P$ comprised between 5 and 35, $pH_{NBS}$ was converted to $pH_T$ according to the electrode response to TRIS buffer solutions prepared at $S_P$ = 5, 15, 25 and 35 and for which the $pH_T$ was assigned at 25°C (Millero, 1986). Reproducibility of pH measurements based on replicate analyses of the same sample or at least two of the three methods used was typically better than ± 0.005.

Dissolved oxygen (DO) concentrations were determined on board by Winkler titration on distinct water samples recovered directly from the Niskin bottles, following the method described by Grasshoff et al. (1999). The relative standard deviation, based on replicate analyses of samples recovered from the same Niskin bottle, was 0.5 %. These measurements served to calibrate the SBE-43 oxygen probe mounted on the rosette sampler.

The stable oxygen isotopic composition of the water samples ($\delta^{18}O_{water}$) was determined using the $CO_2$ equilibration method of Epstein and Mayeda (1953). Aliquots (200 µL) of the water samples and three laboratory internal reference waters were transferred into 3 mL vials stoppered with a septum cap. The vials were then placed in a heated rack maintained at 40°C. Commercially available 99.998% pure $CO_2$ gas (Research Grade) was introduced in all the vials using a Micromass AquaPrep and allowed to equilibrate for 7 hours. The headspace $CO_2$ was then sampled by the Micromass AquaPrep, dried on a -80°C water trap, and analyzed on a Micromass Isoprime universal triple collector isotope ratio mass spectrometer in dual inlet mode at the GEOTOP-UQAM Stable Isotope Laboratory. Data were normalized against the three internal reference waters, themselves calibrated against Vienna Standard Mean Ocean Water (V-SMOW) and Vienna Standard Light Arctic Precipitation (V-SLAP). The results are reported on the δ-scale in ‰ relative to V-SMOW:

$$\delta^{18}O = \left(\frac{(^{18}O/^{16}O)_{sample}}{(^{18}O/^{16}O)_{standard}} - 1\right) \times 1000 \tag{1}$$

Based on replicate analyses of the samples, the average standard deviation of the measurements was better than 0.05‰.

TA was measured using an automated Radiometer (TitraLab865®) potentiometric titrator and a Red Rod® combination pH electrode (pHC2001) at McGill University. The dilute HCl titrant was calibrated prior, during and after each titration session using certified reference materials (CRM) provided by Andrew Dickson (Scripps Institution of Oceanography). Raw titration data were processed with a proprietary algorithm designed for shallow end-point detection. Surface water samples from the Saguenay Fjord and the Upper St. Lawrence Estuary were also analyzed at Dalhousie University using a VINDTA 3C® (Versatile Instrument for the Determination of Titration Alkalinity, by Marianda) following the method described in Dickson et al. (2007). A calibration of the instrument was performed against CRMs and the reproducibility of the measurements was better than 0.1%.

The DIC concentration of samples, recovered in 2016, 2017 and 2018 in the Saguenay Fjord and surface waters of the Upper and Lower St-Lawrence Estuary, were determined at Dalhousie University using the VINDTA 3C®. In 2014, DIC was determined on board using a SciTech Apollo DIC analyzer. Once thermally equilibrated at 25°C, 1-1.5 mL of the sample was acidified with 10% $H_3PO_4$ after being injected into the instrument's reactor. The

215 evolved $CO_2$ was carried to a LI-COR infrared analyzer by a stream of pure nitrogen. A calibration curve was constructed using gravimetrically prepared $Na_2CO_3$ solutions, and the accuracy of the measurements was verified using a CRM. Reproducibility was typically on the order of 0.2%.

**2.4 Calculations**

**2.4.1 Water mass distribution analysis**

A combination of transport processes associated with ocean circulation and biogeochemical cycles generally controls the distribution of tracers in the ocean (Chester, 1990). Resolving the effects of mixing and biogeochemical cycling is imperative if one is to evaluate the movement of nutrients and tracers in a water body. An Optimum Multi-Parameter (OMP) analysis allows for the determination of the relative contributions of pre-defined source-water types (SWT), representing the parameter values of the unmixed water masses in one specific geographic location, by

optimizing the hydrographic data gathered in a given system (Tomczak, 1981). The original OMP algorithm is a linear inverse model that assumes all hydrographic tracers are conservative. The algorithm has since been modified to handle non-conservative properties such as DIC and nutrients by taking into consideration the stoichiometry of microbial respiration and photosynthesis (Dinauer and Mucci, 2018; Karstensen and Tomczak, 1998).

   OMP calculates the SWT fractions, $x_i$, for each data point by finding the best linear mixing combination

defined by parameters such as T, $S_P$, $\delta^{18}O_{water}$, DO, TA, and DIC. The contributions from all SWT must add-up to 100% and cannot be negative. Assuming that four SWT (a, b, c, and d) are sufficient to characterize the water column structure, and six parameters (T, S, $\delta^{18}O_{water}$, DO, TA, and DIC) characterize each of these, the following set of linear equations is solved in the classical OMP analysis (MATLAB - version 1.2.0.0; Karstensen, 2013):

$$x_a T_a + x_b T_b + x_c T_c + x_d T_d = T_{obs} + R_T \tag{2.a}$$
$$x_a S_a + x_b S_b + x_c S_c + x_d S_d = S_{obs} + R_S \tag{2.b}$$
$$x_a \delta^{18}O_a + x_b \delta^{18}O_b + x_c \delta^{18}O_c + x_d \delta^{18}O_d = \delta^{18}O_{obs} + R_{\delta^{18}O} \tag{2.c}$$
$$x_a DO_a + x_b DO_b + x_c DO_c + x_d DO_d = DO_{obs} + R_{DO} \tag{2.d}$$
$$x_a TA_a + x_b TA_b + x_c TA_c + x_d TA_d = TA_{obs} + R_{TA} \tag{2.e}$$
$$x_a DIC_a + x_b DIC_b + x_c DIC_c + x_d DIC_d = DIC_{obs} + R_{DIC} \tag{2.f}$$
$$x_a + x_b + x_c + x_d = 1 + R_\Sigma \tag{2.g}$$

   where $T_{obs}, S_{obs}, \delta^{18}O_{obs}, DO_{obs}, TA_{obs},$ and $DIC_{obs}$ are the observed values in any given parcel of water and $R$ are their respective associated fitting residuals. $T_i,\ S_i,\ \delta^{18}O_i, DO_i, TA_i,$ and $DIC_i$ (i= a, …, d) are the characteristic values

of each SWT (Lansard et al., 2012; Tomczak and Large, 1989; Mackas et al., 1987). Mass conservation is expressed in Eq. (2.g).

   To account for potential environmental variability, measurement inaccuracies, and allow for the comparison of parameters with incommensurable units, a weighting procedure based on covariances between tracers is applied. In this study, weights were assigned arbitrarily based on their conservative behaviors and variability (Lansard et al.,

2012). Conservative tracers (i.e. $S_P$, TA, $\delta^{18}O_{water}$) were assigned heavy weights, while non-conservative tracers (i.e. T, DO, DIC) were given low weights according to their seasonal variability. For instance, temperatures in the surface waters of the Saguenay River range from 3.1°C in the winter to 21°C in the summer. Dissolved oxygen was also considered a non-conservative tracer as it is heavily reliant on temperature and salinity, as well as biological activity. DIC was given an intermediate weight given that it is relatively conservative except in the surface waters, where photosynthesis and air-sea gas exchange take place. Several OMP analyses were carried out using different weights for each parameter, while weighing their conservative behaviour appropriately (i.e., highly conservative vs. lightly conservative). Results were not affected significantly.

### 2.4.2 Source-Water Type definitions

A water mass is, by definition, a body of water having its origin in a particular source region (Tomczak, 1999). An OMP analysis requires the user to define the major water masses contributing to the structure of the water column in the study area. In the context of biogeochemical cycles, a SWT should be defined where the water mass enters the basin, before it enters the mixing region (Karstensen, 2013). Parameter values are preferably extrapolated from hydrographic observations in the water mass formation region or can be found in the literature.

In this study, source-water type definitions were derived from property-property diagrams (See Appendix, Fig. A) of an observational dataset relevant to the Saguenay Fjord: the Saguenay River (SWR), the St. Lawrence Estuary summertime Cold Intermediate Layer (CIL), the Lower St. Lawrence Estuary bottom waters (LSLE) and the St. Lawrence River (SLRW). Each definition was captured relative to the fjord, i.e. each source-water type is only appropriate for the fjord and for the period of study. Definitions and weights are reported in Table 1. A seasonality analysis was carried out to ensure SWT definitions were appropriate for the period of study. Insignificant variations were observed in tracers such as $\delta^{18}O$, DIC, TA, DO and $S_P$. The only highly variable tracer was T, which was given the lowest possible weight in the OMP analysis.

### 2.4.3 CO₂ partial pressures

The $CO_2$ partial pressure in seawater ($pCO_{2(SW)}$) is defined as the $pCO_2$ in water-saturated air ($pCO_{2(air)}$) in equilibrium with the water sample or the ratio of the $CO_2$ concentration in solution to the equilibrium concentration at T, P and $S_P$, multiplied by the actual $pCO_{2(air)}$. As direct measurements of the surface mixed layer $pCO_2$ were not available in September 2014, May 2016, June 2017 and November 2017, it was calculated ($pCO_{2(SW-calc)}$) using CO2SYS (Excel v2.1; Pierrot et al., 2006) and the measured pH (total or NBS/NIST scale; see Appendix, Fig. B.1 and B.2), DIC ($\mu mol \cdot kg^{-1}$), in-situ temperature (°C), practical salinity ($S_P$) and pressure (dbar) as input parameters. When available, soluble reactive phosphate (SRP) and dissolved silicate (DSi) concentrations were also included in the calculations, but their inclusion did not affect the results significantly because their concentrations are relatively low in surface waters (0.49 μM and 37.0 μM, respectively) and introduce an insignificant error. DIC rather than TA was used as an input parameter to CO2SYS since the fjord surface waters are enriched in colored dissolved organic carbon (> 4 mg/L) delivered by the Saguenay River, and are characterized by a negative organic alkalinity (positive organic acidity) (see below). The carbonic acid dissociation constants ($K^*_1$ and $K^*_2$) of Cai and Wang (1998) were

used for the calculations, as the latter were found to be more suitable for the low-salinity waters encountered in estuarine environments such as the Saguenay Fjord ($S_P < 20$) (Dinauer and Mucci, 2017). $pCO_{2(SW-calc)}$ values were computed for the surface mixed layer located above the sharp pycnocline (~10 m) where most physical and chemical properties are directly impacted by biological activity (photosynthesis and respiration) as well as heat and gas exchange across the air-sea interface (Table 2). Direct measurements of $pCO_2$ ($pCO_{2(SW-meas)}$) were acquired in May 2018, and $pCO_{2(SW-calc)}$ were also calculated from pH and DIC for this sampling month for comparison purposes, following the aforementioned procedure.

### 2.4.4 CO$_2$ flux across the air-sea interface

The difference between the air and sea-surface $pCO_2$ values ($\Delta pCO_2 = pCO_{2(SW)} - pCO_{2(air)}$) determines the direction of gas exchange and whether the surface mixed layer of a body of water is a source or a sink of $CO_2$ for the atmosphere. The air-sea $CO_2$ gas exchange, or $CO_2$ flux, can be estimated at each station using the following relationship:

$$FCO_2 = k \cdot K_0 \cdot (\Delta pCO_2)$$ (3.a)

where $F$ is the flux of $CO_2$ across the air-sea interface in mmol·m$^{-2}$·d$^{-1}$, $k$ is the gas transfer velocity of $CO_2$ in cm·h$^{-1}$ (Wanninkhof, 1992), $K_0$ is the solubility of $CO_2$ in mol·kg$^{-1}$·atm$^{-1}$ at the in-situ temperature and salinity of the surface waters (Weiss, 1974), and $\Delta pCO_2$ is the difference between the air and sea-surface $pCO_2$ values in µatm. Whereas, formally, Fick's first law of diffusion should be written as $F = -D \ \delta C/\delta x$ (where $F$ is the diffusion flux in mole sec$^{-1}$ m$^{-2}$, $D$ is the diffusion coefficient in m$^2$ sec$^{-1}$, $C$ is the concentration of $CO_2$ in mole m$^{-3}$ and $x$ is the distance in m), as commonly expressed by Eq. (3.a), positive values of $F$ indicate the release of $CO_2$ to the atmosphere by surface waters, whereas negative values imply that surface waters serve as a sink of atmospheric $CO_2$. The flux of $CO_2$ was computed for each sampling month, using the $pCO_{2(air)}$ for each sampling date (395 µatm for September 2014, 407 µatm for May 2016, 408 µatm for June and November 2017, and 411 µatm for May 2018 – see below for details).

The gas transfer velocity of $CO_2$ was calculated using the revised relationship of Wanninkhof (2014):

$$k = 0.215u^2 \ (Sc/660)^{-1/2}$$ (3.b)

where $u$ is the wind speed (m s$^{-1}$) and $Sc$ is the Schmidt number (Wanninkhof, 2014). Wind speed was estimated using the hourly station wind speed data from Environment Canada at the La Baie weather station (Fig. 1.a), for each sampling month. The Schmidt number is defined as the kinematic viscosity of water divided by the diffusion coefficient of $CO_2$. $Sc$ was corrected for the temperature dependence of $CO_2$ in freshwater ($S_P = 0$), assuming that $k$ is proportional to $Sc^{-1/2}$ (Wanninkhof, 1992). In the case of $CO_2$, the increase in $Sc^{-1/2}$ (and $k$) with increasing temperature is compensated for by a decrease in solubility, therefore $k$ was considered nearly temperature independent (Wanninkhof, 1992). $Sc$ was computed using:

$$Sc = A + Bt + Ct^2 + Dt^3 + Et^4 \tag{3.c}$$

where $t$ is the temperature (degrees Celcius) and $A$, $B$, $C$, $D$ and $E$ are fitting coefficients for seawater ($S_P = 35$) and freshwater ($S_P = 0$), for water temperatures ranging from -2°C to 40°C (Wanninkhof, 2014). The uncertainty in $Sc$ ranges from 3 to 10% and is mainly due to the imprecision of diffusion coefficients (Wanninkhof, 2014). Estimates of $k$, calculated at each sampling point using the equation of Wanninkhof (2014), ranged from 0.36 to 3.38 cm·h$^{-1}$ for the fjord, compared to 1.6 to 4.5 cm·h$^{-1}$ in the St. Lawrence Estuary (Dinauer and Mucci, 2017).

Atmospheric pCO$_2$ values (pCO$_{2(air)}$) were computed using the daily averages of measured mole fractions of CO$_2$ in dry air, obtained at the La Baie weather station and retrieved from the Climate Research Division at Environment and Climate Change Canada. The mean pCO$_{2(air)}$ was then calculated for each year using the following equation:

$$pCO_{2(air)} = xCO_2 \cdot (P_b - P_w) \tag{4}$$

where $xCO_2$ is the measured mole fraction of CO$_2$ in dry air in ppm, $P_b$ is the barometric pressure at the sea surface in atm, and $P_w$ is the saturation water vapor pressure at in-situ temperature and salinity, in atm. $P_b$ was obtained using the conversion formula of Tim Brice and Todd Hall (from NOAA's National Weather Service - https://www.weather.gov/epz/wxcalc_wxcalc2go), using the La Baie weather station's elevation (152 m). $P_w$ was calculated using the Rivière-à-Mars properties (i.e. closest body of water to the weather station) and the $P_w$ calculated from its relationship to T and $S_P$ provided by Weiss and Price (1980).

The area-averaged CO$_2$ flux (F$_{area-avg}$) was computed for the whole fjord, following the procedure described by Jiang et al. (2008):

$$F_{area-avg} = \frac{\Sigma F_i \times S_i}{\Sigma S_i} \tag{5}$$

where $F_i$ is the average of all the fluxes within segment $i$, and $S_i$ is the surface area of segment $i$. The fjord was divided into two segments, one including the inner basin and the other encompassing the two outer basins, as each segment often displays distinct behaviors. Segments are identified in Fig. 1.b. The fjord's surface area (~290 km$^2$) was computed using a land mask in MATLAB.

**2.4.5 Water Mixing Model**

A two end-member mixing model was constructed based on the chemical properties of the freshwater delivered to the fjord (Saguenay River) and marine bottom waters entering the fjord from the St. Lawrence Estuary (Fig. 2.a). As shown in the results of the OMP analysis (Sect. 3.1), the LSLE and SLRW have a negligible influence on the fjord's water structure, and thus were not included in the model. Given that the carbonate chemistries of the CIL and LSLE waters are similar, the bottom waters were assumed to be well mixed and constitute a single end-member. This is illustrated in Fig. (2), as the high $S_P$ end-member alkalinity extends linearly beyond that of the CIL end-member

(Table 1). The measured surface TAs were strongly correlated to $S_P$ ($R^2 = 0.99$) in the fjord waters. Therefore, end-member properties were obtained by extrapolating the surface water (above the pycnocline) data to $S_P = 0$ and bottom-water data to the highest measured salinity (Fig. 2.a). The extrapolated $TA_{(meas)}$ (Fig 2.b; 154 µmol·kg$^{-1}$) is in good agreement with the average $TA_{(meas)}$ of samples taken directly from the Saguenay River in 2013 and 2017 (157 µmol·kg$^{-1}$). The organic alkalinity of the fjord waters was estimated from the difference between the measured and calculated TA ($TA_{(calc)}$; Fig. 2.b). The latter was calculated using CO2SYS (Excel v2.1; Pierrot et al., 2006) and pH and DIC as input parameters. The end-member source waters were then mixed, assuming that $TA_{(calc)}$ and DIC behave conservatively. Hence, the salinity, total alkalinity ($TA_{(mix)}$) and dissolved inorganic carbon ($DIC_{(mix)}$) of the mixed solutions were calculated using the following equations:

$$S_{P(mix)} = \frac{m_1 S_{P1} + m_2 S_{P2}}{(m_1 + m_2)} \tag{6a}$$

$$TA_{(mix)} = \frac{m_1 TA_{(calc)-1} + m_2 TA_{(calc)-2}}{(m_1 + m_2)} \tag{6b}$$

$$DIC_{(mix)} = \frac{m_1 DIC_1 + m_2 DIC_2}{(m_1 + m_2)} \tag{6c}$$

where $m_i$ are the mass contributions of each end-member to the mixture.

$pCO_{2(SW-mix)}$ was then computed from $TA_{(mix)}$ and $DIC_{(mix)}$ for practical salinities ranging from 0 to 33, at four different temperatures (0°C, 5°C, 10°C and 15°C) using CO2SYS. Results of the model (Fig. 8) show that, at the lower and higher salinities, the $pCO_{2(SW-mix)}$ is elevated, and the fjord serves as a net source of $CO_2$ to the atmosphere, but at intermediate salinities ($5 < S_P < 15$) or mixing ratios, the fjord may serve as a net sink of atmospheric $CO_2$ when surface water temperatures are close to freezing. The data from the various cruises are superimposed on the model results, after correction for the organic alkalinity.

### 2.4.6 Salinity normalization of DIC in surface waters

To quantitatively evaluate the impact of biological activity on the DIC budget in the surface waters of the fjord, DIC and $TA_{(calc)}$ were normalized to the average surface salinity of each sampling month ($S_P = 12.4$ for September 2014, $S_P = 2.58$ for May 2016, $S_P = 7.61$ for June 2017, $S_P = 10.9$ for November 2017 and $S_P = 5.9$ for May 2018) following the procedure of Friis et al. (2003):

$$\text{NDIC} = \frac{DIC^{meas} - DIC^{S=0}}{S^{meas}} \cdot S^{ref} + DIC^{S=0} \tag{7}$$

where $DIC^{meas}$ is the measured DIC, $DIC^{S=0}$ is the DIC extrapolated to $S_P = 0$, $S^{meas}$ is the measured practical salinity and $S^{ref}$ is the average measured practical salinity per sampling month (Friis et al., 2003). The change in NDIC (i.e. ΔNDIC) along the fjord, relative to the waters at the head of the fjord, was then computed for each sampling month. These values reveal how DIC evolves along the fjord beyond what is expected based on conservative mixing.

### 2.4.7 Oxygen saturation and apparent oxygen utilization in the surface waters

To further account for the biological activity in the surface waters, the oxygen saturation index was calculated for each sampling month in the surface waters of the fjord using:

$$\% \, sat = \left([O_2]_{meas}/[O_2]_{equil}\right) \times 100 \tag{8}$$

where $[O_2]_{meas}$ is the dissolved oxygen concentration measured in the fjord waters, and $[O_2]_{equil}$ is the equilibrium dissolved oxygen concentration (or solubility) at in-situ conditions (i.e. temperature and salinity) for each sample.

The oxygen saturation index indicates if the system is autotrophic (i.e. production of oxygen, dominated by photosynthesis) or heterotrophic (consumption of oxygen, dominated by microbial respiration). The oxygen saturation remains a qualitative proxy as $O_2$ exchange at the air-sea interface is about 9 times faster than $CO_2$ exchange (Zeebe and Wolf-Gladrow, 2001). The apparent oxygen utilization (AOU) was also computed from the difference between $[O_2]_{equil}$ and $[O_2]_{meas}$.

## 3 Results and discussion

### 3.1 Water mass analysis

Relative contributions (mixing ratios, f) of the Saguenay River (SRW), the St. Lawrence Estuary summertime Cold Intermediate Layer (CIL) and Lower St. Lawrence Estuary (LSLE) bottom waters throughout the Saguenay Fjord's water column for the sampling month of June 2017 are shown in Fig. (3). As expected, the SRW and CIL are dominant contributors, with the SRW forming a brackish surface layer (f = 1 in surface waters), and the CIL replenishing the bottom waters of the fjord (0.7 < f < 1). According to the OMP analysis, the LSLE bottom waters have a small contribution to the fjord's bottom waters (f = 0.2), adding to the complexity of the water structure. Although somewhat unexpected, this can readily be explained by tidal upwelling, internal waves and intense turbulent mixing of the water column resulting from the rapid shoaling at the head of the Laurentian Channel (Gratton et al., 1988; Saucier and Chassé, 2000). The relative contribution of the LSLE bottom waters in the deep waters of the fjord is small and could only be detected because of the suite of geochemical and isotopic tracers used in the OMP analysis, especially the difference in the $\delta^{18}O_{water}$ signature of the CIL and LSLE waters. The contribution from the St. Lawrence River Water (SLRW) is negligible, as it intrudes slightly at the surface at the mouth of the fjord and is thus not shown here. Although the water column structure is similar throughout the year, seasonal variations do occur and will be addressed in a forthcoming paper.

### 3.2 Aqueous pCO₂ and CO₂ flux

Variations of the inorganic carbon chemistry in the Saguenay Fjord water column are described using field data acquired in September 2014, May 2016, June 2017, November 2017 and May 2018. The organic alkalinity (acidity) accounted, on average, for 2.2% to 11.9% of the total alkalinity of the Saguenay River and varied annually and seasonally (-21 µmol·kg⁻¹ in September 2014, -39 µmol·kg⁻¹ in May 2016, -49 µmol·kg⁻¹ in June 2017, -22

μmol·kg$^{-1}$ in November 2017 and -18 μmol·kg$^{-1}$ in May 2018). It was inversely proportional to the salinity of the surface waters of the fjord and became positive yet a negligible fraction (< 0.1%) of TA$_{(corr)}$ at S$_P$ > 25, like in the St. Lawrence Estuary. The negative organic alkalinity of the Saguenay River water most likely originates from soil humic acids that are flushed by percolation with groundwaters that drain the metamorphic and igneous rocks of the Canadian Shield. Surface-water pCO$_2$ (pCO$_{2(SW-calc)}$) values were higher at the head of the fjord (i.e. near the Saguenay River mouth) and lower at the mouth of the fjord, although large variations (315 μatm to 740 μatm – average 503 μatm) were observed on a seasonal and yearly basis (Table 2). Values of pCO$_{2(SW)}$ were higher in May 2018 (623 μatm), June 2017 (506 μatm) and May 2016 (563 μatm) than in November 2017 (418 μatm) and September 2014 (406 μatm). This can be explained by the larger freshwater discharge from the Saguenay River in the spring (i.e. spring freshet, average of 1856 ± 21 m$^3$ s$^{-1}$ for spring periods of 1998 - 2018), compared to the fall (1470 ± 10 m$^3$ s$^{-1}$ for fall periods of 1998 - 2018). As atmospheric pCO$_{2(air)}$ varied marginally between September 2014 (395 μatm) and May 2018 (411 μatm), the fjord was generally a source of CO$_2$ to the atmosphere near its head (i.e. surface pCO$_2$ values above atmospheric level), while the zone near its mouth was most often a sink (i.e. surface pCO$_2$ values below atmospheric level) (Fig. 4). An anomaly was observed in November 2017, with a high pCO$_{2(SW-calc)}$ value (> 550 μatm) near the mouth of the fjord. Given the statistics of the box plot presented in Fig. (7), this value appears to be erroneous.

Air-sea CO$_2$ fluxes within the fjord ranged from -2.4 mmol·m$^{-2}$·d$^{-1}$ to 10.0 mmol·m$^{-2}$·d$^{-1}$ (Fig. 6). Near the head of the fjord, fluxes were mostly positive, while values decreased when approaching its mouth. Overall, the total area-averaged degassing flux of the fjord adds up to 2.14 ± 0.43 mmol·m$^{-2}$·d$^{-1}$ or 0.78 ± 0.16 mol·m$^{-2}$·yr$^{-1}$. In comparison, the degassing flux in the adjacent St. Lawrence Estuary was estimated at between 0.36 and 0.74 mol·m$^{-2}$·yr$^{-1}$ during the late spring and early summer (Dinauer and Mucci, 2017). This discrepancy can be explained by the low carbonate alkalinity (and buffer capacity) of the Saguenay River waters that flow through the Grenvillian metamorphic and igneous rocks of the Canadian Shield (Piper et al., 1990), as with most rivers on the north shore of the St. Lawrence Estuary (e.g., Betsiamites, Manicouagan, Romaine; Paul del Giorgio, pers. comm.), and the low productivity of the fjord surface waters because of very limited light penetration due to their high chromotrophic dissolved organic matter (CDOM) content (Tremblay and Gagné, 2009; Xie et al., 2012). In contrast, waters of the St. Lawrence River have an elevated carbonate alkalinity (~1200 μM), inherited from the Ottawa River that drains through limestone deposits (Telmer and Veizer, 1999). Furthermore, the Estuary is host to multiple seasonal phytoplankton blooms (Levasseur and Therriault, 1987; Zakardjian et al., 2000; Annane et al., 2015) that strongly modulate its trophic status (Dinauer and Mucci, 2018).

The correlation between pCO$_{2(SW-meas)}$ and pCO$_{2(SW-calc)}$ is presented in Fig. (5). The average difference between pCO$_{2(SW-meas)}$ and pCO$_{2(SW-calc)}$ is 48 μatm, implying that calculations underestimate pCO$_{2(SW)}$ values by approximately 7% and thus contribute to the uncertainty associated with CO$_2$ fluxes. This discrepancy most likely originates from uncertainties associated with the carbonic acid dissociation constants (K*$_1$ and K*$_2$) in low salinity estuarine environments, particularly those affected by strong organic alkalinities or acidities such as in the Saguenay Fjord (Cai et al., 1998; Ko et al., 2016). This concurs with the results of Lueker et al. (2000) who showed that, depending on the choice of K*$_1$ and K*$_2$, computed pCO$_{2(SW)}$ values from other carbonate system parameters (TA, DIC, pH) can be up to 10% lower than those of direct measurements. Consequently, although the constants of Cai and Wang (1998) are

the most suitable for this study, direct measurements of the $pCO_{2(SW)}$ should preferentially be carried out whenever possible.

### 3.3 Water Mixing Model approach

As results of the OMP analysis reveal, LSLE and SLRW have a negligible influence on the water properties in the fjord, except for the latter near the mouth. Additionally, given the relatively small contribution of the LSLE deep waters and their similarity to the carbonate chemistry of the CIL, their influence is considered inconsequential on the properties of the mixture. Hence, a conservative mixing model was constructed based on the chemical properties of the two main source-water masses in the fjord (i.e., SRW and the CIL mixture for bottom waters), and the relationship between practical salinity and $TA_{(corr)}$/DIC, respectively (Fig. 8). $pCO_{2(SW-calc)}$ were normalized at each station to the average surface water temperature per sampling month (i.e., T = 10.4°C for September 2014, T = 5.04°C for May 2016, T = 11.9°C for June 2017, T = 7.13°C for November 2017 and T = 5.08°C for May 2018) to account for the effects of temperature on the $CO_2$ solubility in water, following the procedure described in Jiang et al. (2008). The temperature-normalized $pCO_{2(SW-calc)}$ values, $pCO_{2(SW-SST)}$, from the various cruises were superimposed on the model results in Fig. (8).

Field measurements follow the trend displayed by the mixing model. The fjord appears to be a net source of $CO_2$ to the atmosphere during periods of high freshwater discharge (i.e. spring freshet) and a net sink at intermediate surface salinities ($5 < S_P < 15$). This is consistent with the weak buffer capacity of the freshwater. Given the short residence time of surface waters in the Saguenay Fjord (~ 1.5 days), the influence of gas exchange across the air-sea interface is negligible on the DIC pool. Likewise, Dinauer and Mucci (2017) reported that the surface waters in the St. Lawrence Estuary near Tadoussac (at the mouth of the fjord) are highly supersaturated in $CO_2$ with respect to the atmosphere and only the highly productive waters of the Lower Estuary manage to draw down the surface $pCO_2$ to near atmospheric values. In other words, degassing of the metabolic $CO_2$ accumulated in the river and upper estuary is slow. Thus, changes in temperature-normalized $pCO_2$ primarily reflect changes in DIC by mixing and biological activity. Hence, discrepancies between results of the mixing model and field measurements can be ascribed to microbial respiration and photosynthesis.

In May 2016, the surface waters of the fjord were clearly supersaturated in oxygen (Fig. 9), implying that photosynthesis dominated over respiration. This would explain the rapid seaward (increasing $S_P$) decrease in $pCO_{2(SW-SST)}$, faster than the mixing model predicts (Fig. 8), and the strong negative ΔNDIC (i.e. change in NDIC relative to the saline waters at the head of the fjord) throughout the fjord (Fig. 10), as $CO_2$ (i.e. DIC) is taken up by photosynthesizing organisms - most likely diatoms (Chassé and Côté, 1991). In May 2018, surface waters were slightly undersaturated in oxygen, between 90% and 100% saturation, and ΔNDIC was positive over most of the fjord. Very similar trends were observed in June 2017, with near-saturation oxygen concentrations (between 95 and 101% saturation) and mostly positive ΔNDIC values throughout the transect. Thus, during these sampling periods, biological activity was dominated by microbial respiration (Fig. 10), elucidating the minor deviation between the $pCO_{2(SW-SST)}$ and the model results (Fig. 8), especially near the head of the fjord. Additionally, it is interesting to note that ΔNDIC is chronically negative for all sampling months near the 45 km mark.

The difference between the May 2016 and May 2018 biological responses to the spring freshet can potentially be explained by the difference in total freshwater discharge from the Saguenay River to the fjord. The freshwater discharge in May 2018 was approximately 20% larger than in May 2016. Whereas the surface salinities recorded both years throughout most of the fjord were nearly identical ($S_P = 0.5$-4 in 2016, $S_P = 0.7$-5 in 2018), the greater delivery of soil porewater and associated CDOM in May 2018 may have inhibited local productivity due to light absorption by CDOM (Lavoie et al., 2007). Consistent with this interpretation is the fact that the $pCO_{2(SW-SST)}$ at the head of the fjord (St. Fulgence) in May 2016 was slightly higher than in May 2018 but was drawn down much faster downstream (Fig. 8).

There does not appear to be a clear biological signal in the November 2017 data, as little variation is observed between the measured and modeled $pCO_2$. Furthermore, Fig. (10) indicates that neither respiration nor photosynthesis dominated during this period as the $\Delta$NDIC varies between weakly positive and negative values. Likewise, the September 2014 data reveal little biological activity, although a slight dominance of respiration (i.e. positive $\Delta$NDIC with a drop in %$O_2$ saturation) can be observed near the mouth of the fjord (Fig. 10), hence explaining the slight deviation from the mixing model.

These results highlight the importance of the freshwater plume from the Saguenay River in regulating the $pCO_2$ dynamics in the fjord. Winds, in addition to regulating the gas exchange coefficient, are also known to have a direct influence on air-sea $CO_2$ fluxes by driving upwelling of $CO_2$-rich waters along with the entrainment of nutrients in surface waters, thus increasing biological activity (Wanninkhof and Triñanes, 2017). However, wind speeds are relatively low in the studied system (1.89 m s$^{-1}$ < u < 4.2 m s$^{-1}$, Table 2), implying a calm sea state (Frankignoulle, 1998), and hence reinforcing that changes in $pCO_{2(SW-SST)}$ can mainly be attributed to microbial respiration and photosynthesis modulated by water renewals rather than winds.

**4 Summary and conclusions**

Results of the OMP analysis reveal that SRW and CIL are the dominant source-water types to the fjord and determine the structure of its water column. Mixing of marine waters with SRW at the head of the fjord leads to the formation of a brackish surface layer (wedge) while the CIL replenishes the bottom waters of the fjord. The analysis further unveiled a small contribution of the LSLE bottom water to the bottom waters of the fjord, adding to the complexity of the water column structure. The SLRW has a negligible influence on the water properties in the fjord, except near its mouth - sampling of the very turbulent waters directly over the fjord's first and shallowest sill would help refine the contribution of the SLRW to the fjord's surface waters.

The magnitude and sign of the $\Delta pCO_2$ across the air-water interface in the Saguenay Fjord, mostly determined by the $pCO_{2(SW)}$ as $pCO_{2(air)}$ varied only slightly over the sampling period, are mostly modulated by the freshwater discharge and the salinity of the surface waters. The surface waters of the fjord are a source of $CO_2$ to the atmosphere at high freshwater discharge, and a sink of $CO_2$ at intermediate surface salinities ($5 < S_P < 15$), especially at near-freezing temperatures. Direct measurements of surface-water $pCO_2$ acquired in May 2018 confirmed this trend. Biological activity alters the surface-water $pCO_2$, with both photosynthesis and respiration impacting the waters

depending on the sampling month. This conclusion is supported by the oxygen saturations observed in the surface waters of the fjord, as well as the downstream $\Delta$NDIC trend along the fjord's main axis. Given the short residence time of surface waters in the Saguenay Fjord (~ 1.5 days), the influence of gas exchange on spatial variations of the $\Delta pCO_2$ across the air-sea interface along the main axis of the fjord is negligible on the DIC pool. Overall, the fjord serves as a source of $CO_2$ to the atmosphere during the ice-free season, with an average yearly outgassing flux of 0.78 $\pm$ 0.16 mol·m$^{-2}$·yr$^{-1}$.

A study of the dissolved inorganic carbon budget of fjords not only affords information on their carbon dioxide levels (i.e. source or sink of $CO_2$ with respect to the atmosphere) and surface-water chemistry, but also provides insights on the magnitude of gas exchange and the amount of biological activity it sustains. In addition to biological production, upwelling, water temperature, and the spreading of freshwater plumes all regulate $pCO_2$ in coastal systems. Wind speed is also critical as it impacts sea state and the efficiency of gas exchange at the air-sea interface (Chen et al., 2013). Hence, the importance of wind on controlling the $CO_2$ flux needs to be further investigated, especially in high latitude fjords where strong winds are often focussed along narrow channels between steep cliffs and narrow inlets.

Anthropogenic activities, through climate change, are altering the continental water cycle, along with the flows of carbon, nutrients and sediment to the coastal oceans (Borges, 2005), and therefore, the sequestration of anthropogenic $CO_2$ by the oceans. In glacial fjords, freshwater discharge will be modified by accelerated glacier melting and their ultimate demise. Knowledge of the spatial and temporal variability of glacial discharge and seasonal ice cover in high latitude fjords is essential to estimate their influence on circulation patterns and freshwater export, both of which impact local productivity, terrestrial carbon burial and export as well as $CO_2$ fluxes in these complex ecosystems. Finally, an improved understanding of the coastal carbon cycle will require a more comprehensive spatial and temporal coverage of surface mixed layer $pCO_2$ data in these environments, ideally from direct measurements.

    **Figures**

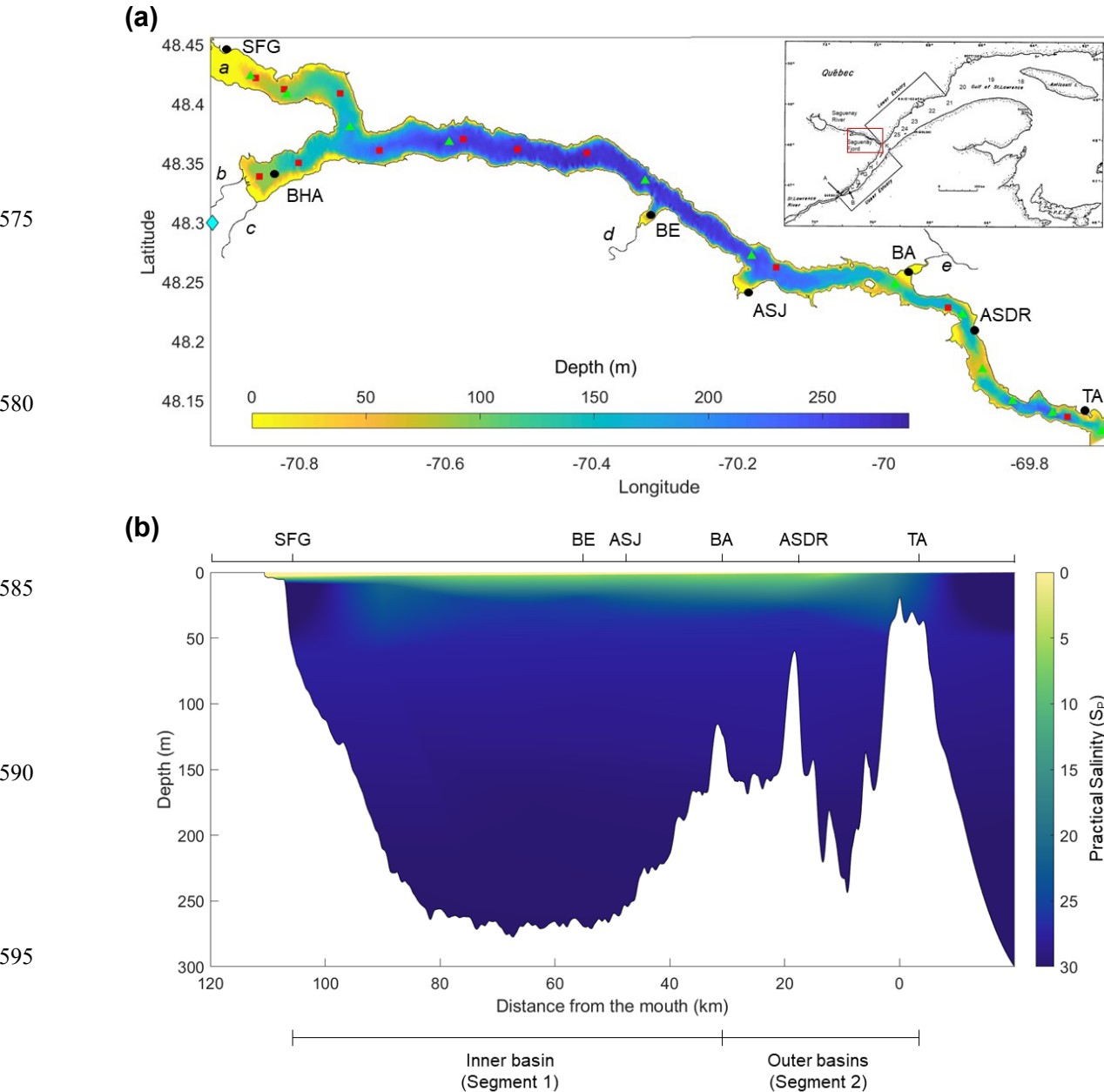

Figure 1. a) Bathymetry and geographic location of the Saguenay Fjord. Red squares represent the hydrographic stations sampled during R/V Coriolis II cruises in September 2014, May 2016, June 2017 and May 2018. Green triangles represent the hydrographic stations sampled during a SECO.net cruise onboard the R/V Coriolis II in November 2017. The approximate locations of the following are shown: Tadoussac (TA), L'Anse de Roche (ASDR), Baie Sainte-Marguerite (BA), Anse-Saint-Jean (ASJ), Baie-Eternité (BE), St. Fulgence (SFG), Baie des Ha! Ha! (BHA). The main tributaries to the fjord are also shown, including the Saguenay River (*a*), Rivière-à-Mars (*b*), Rivière des Ha! Ha! (*c*), Éternité River (*d*) and Sainte-Marguerite River (*e*). The blue diamond identifies the location of the La Baie weather station. Letters (A to K) and numbers (18 to 25) in the inset indicate the location of sampling stations in the St. Lawrence Estuary where data were acquired to define the SLRW, LSLE and CIL end-members. b) Longitudinal section along the Saguenay Fjord, showing the strong halocline.

**(a)**

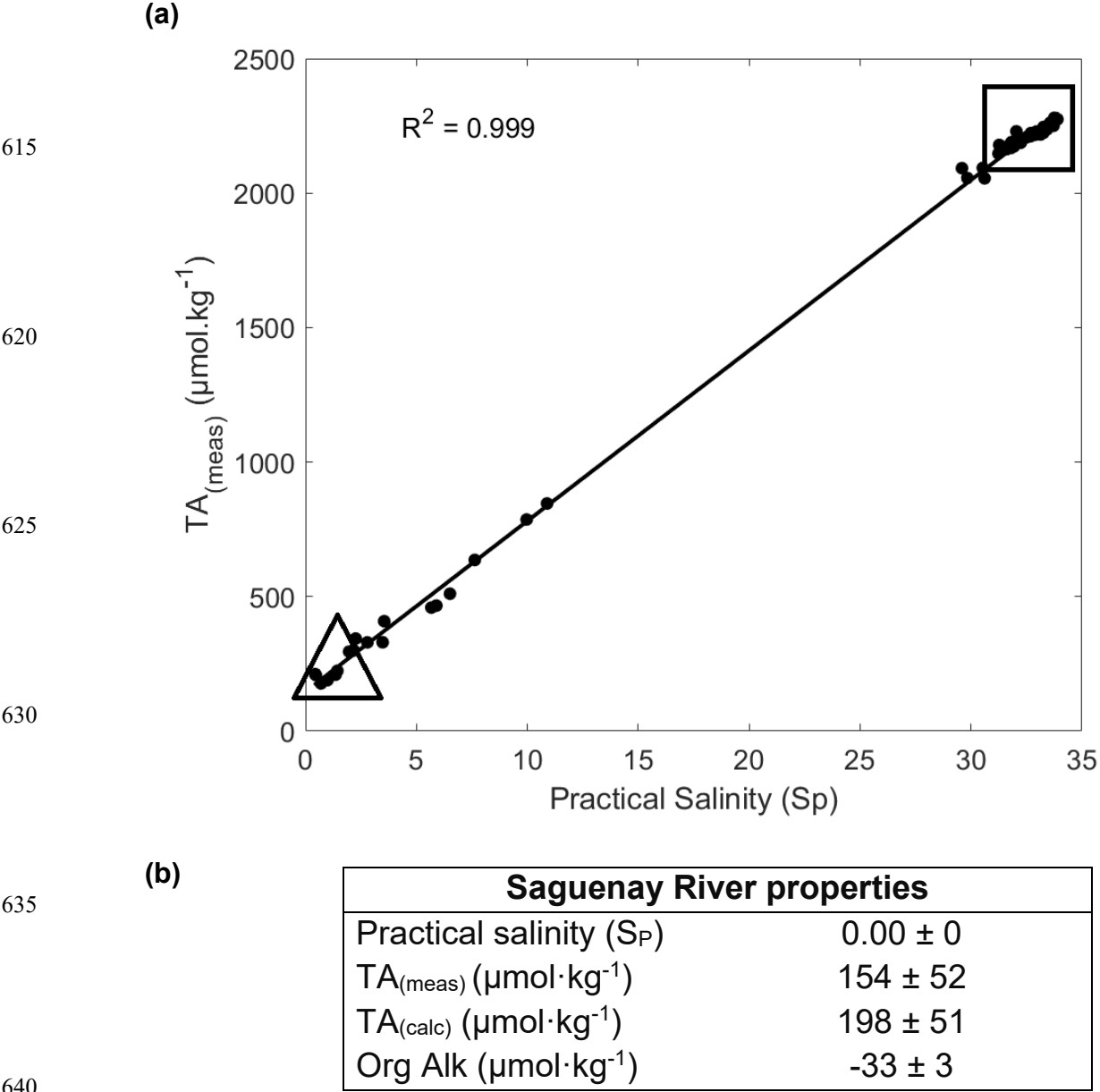

**(b)**

| Saguenay River properties | |
|---|---|
| Practical salinity ($S_P$) | 0.00 ± 0 |
| $TA_{(meas)}$ ($\mu mol \cdot kg^{-1}$) | 154 ± 52 |
| $TA_{(calc)}$ ($\mu mol \cdot kg^{-1}$) | 198 ± 51 |
| Org Alk ($\mu mol \cdot kg^{-1}$) | -33 ± 3 |

**Figure 2. a) Measured alkalinity ($TA_{(meas)}$) versus practical salinity ($S_P$) for SRW and CIL data points, for all sampling months ($R^2$ = 0.999). The triangle defines the properties of the SRW, and the square comprises the properties of the CIL. b) $TA_{(meas)}$, $TA_{(calc)}$ and Org Alk definitions for the Saguenay River (SRW), using surface-water data from all sampling months, with standard error. The Org Alk (positive) contribution to the TA of the CIL is not considered as it accounts for less than 0.1 % of its TA.**

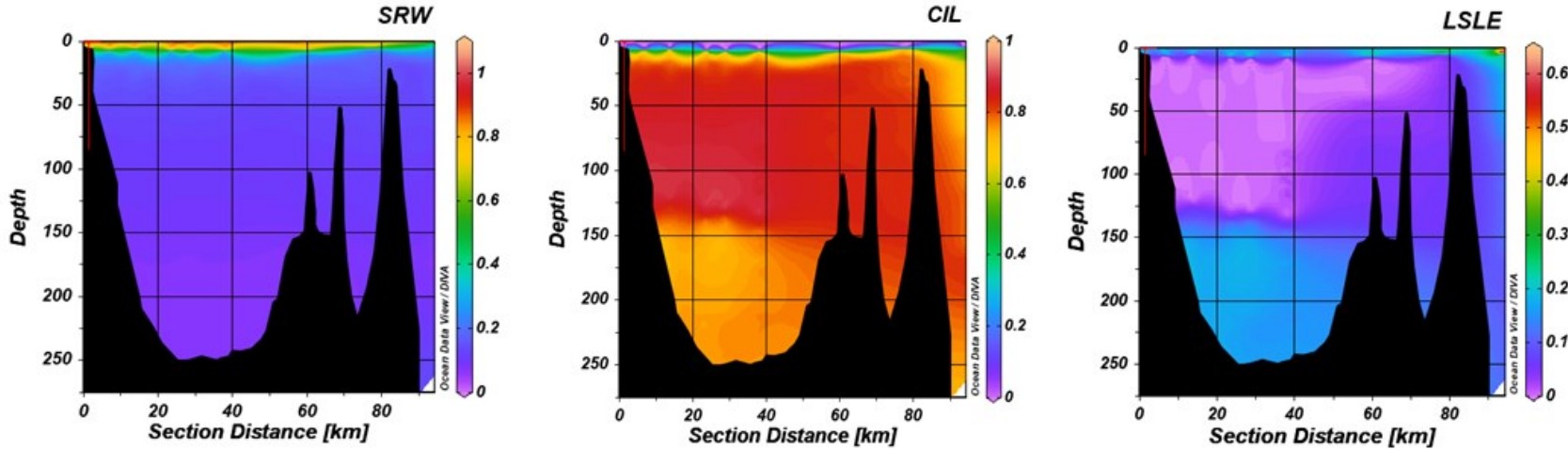

**Figure 3. Vertical sections showing the relative contributions of the Saguenay River (SRW), the St. Lawrence Estuary Cold Intermediate Layer (CIL) and Lower St. Lawrence Estuary bottom waters (LSLE) to the water column structure of the Saguenay Fjord (June 2017). Fractions were estimated using an Optimum Multi-Parameter (OMP) algorithm (Tomczak and Large, 1989; Tomczak, 1981; Mackas and Harrison, 1997). A Variational Analysis (DIVA) interpolation was applied between field data points in Ocean Data View.**

650

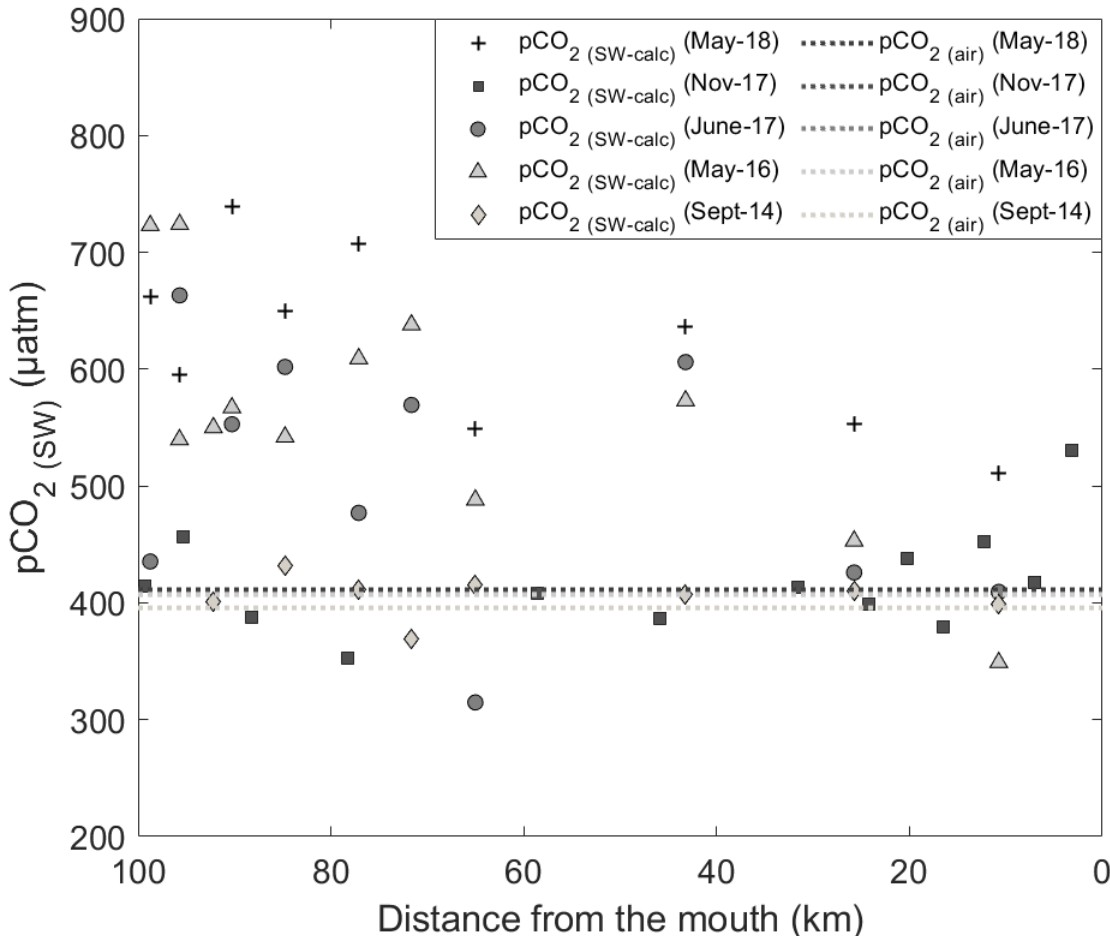

**Figure 4. Spatial distribution of surface-water pCO2(SW-calc) in June 2017, November 2017, May 2016, September 2014 and May 2018. Dashed lines represent the pCO2(air) on the sampling months (respectively 396 ppm in September 2014, 407 ppm in May 2016, 408 ppm in June and November 2017, and 411 ppm in May 2018). Data points above the red line indicate that waters are sources of CO2 to the atmosphere, whereas those below the red line identify waters that are sinks of atmospheric CO2.**

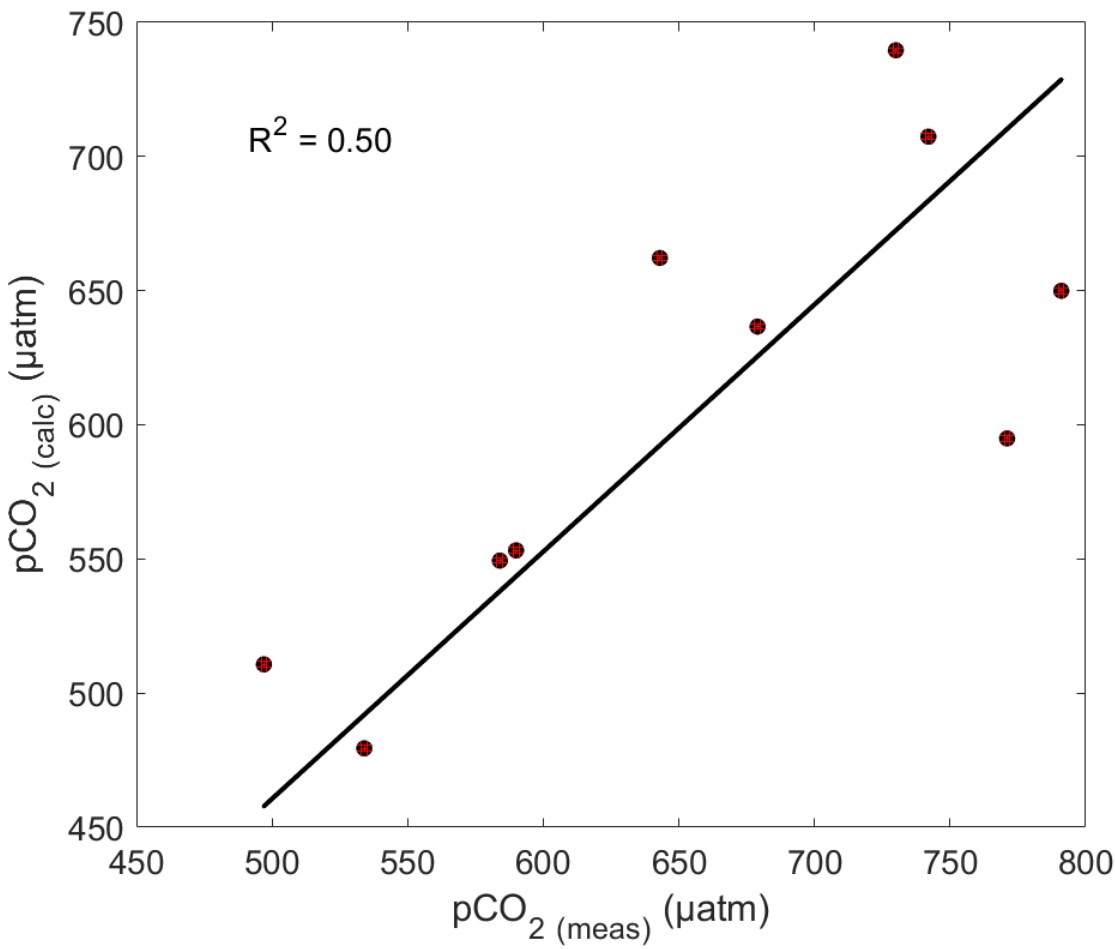

**Figure 5. Correlation between pCO$_{2(SW-meas)}$ and pCO$_{2(SW-calc)}$ for May 2018. The black line shows the linear regression with a null intercept (R$^2$ = 0.50). Error bars, in red, are smaller than the symbol.**

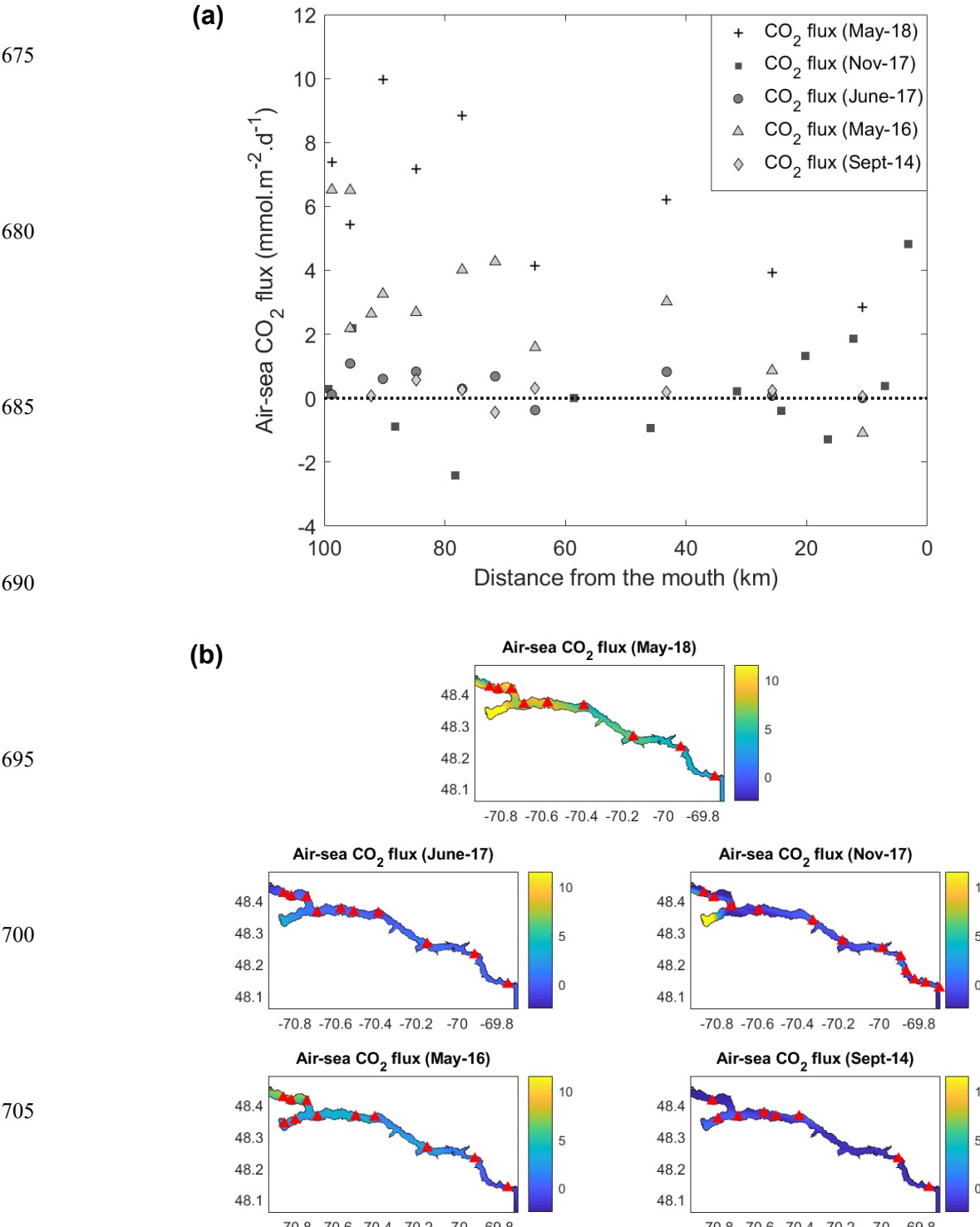

**Figure 6. a) Spatial distribution of air-sea CO₂ flux (mmol·m⁻²·d⁻¹) in the Saguenay Fjord for all cruises. Data points above the dashed line indicate sources of CO₂ to the atmosphere, whereas those below the red line are sinks of atmospheric CO₂; b) Spatial interpolation of air-sea CO₂ fluxes (mmol·m⁻²·d⁻¹) in the Saguenay Fjord for all cruises. Red triangles identify sampling locations.**

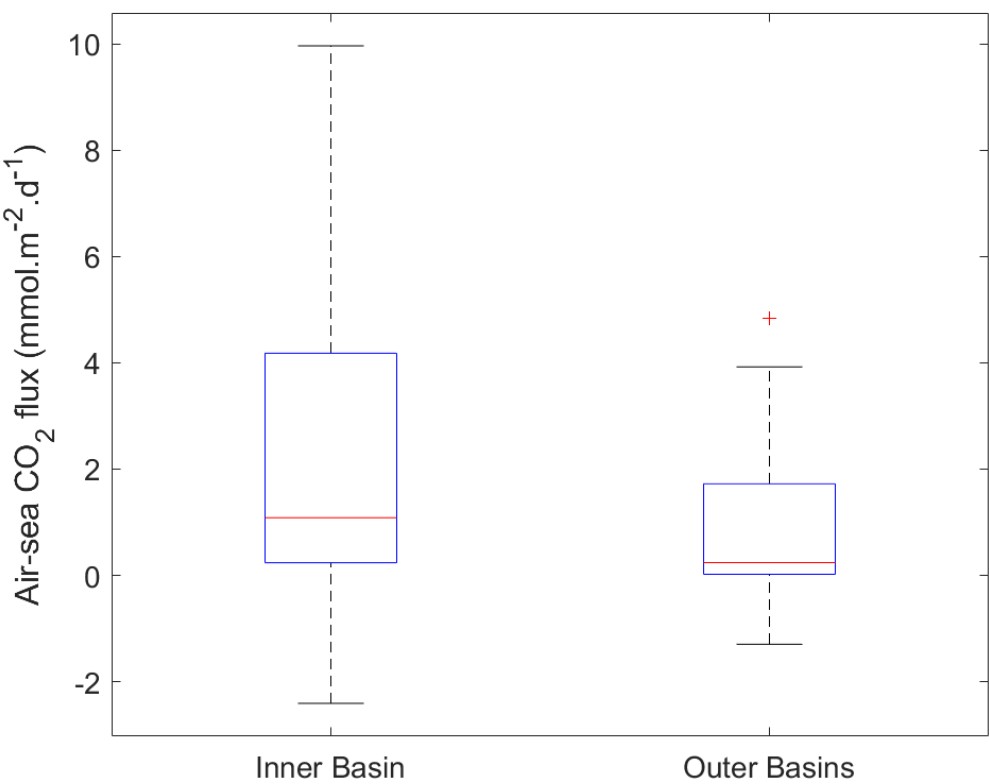

**Figure 7. Box-plot of the air-sea CO₂ fluxes from all data in the two subsections of the study area (Inner basin and Outer basins). The red line is the median, the box spans the interquartile range (25-75 percentiles) and the whiskers show the extreme data points not considered outliers. One outlier is identified by the red + symbol.**

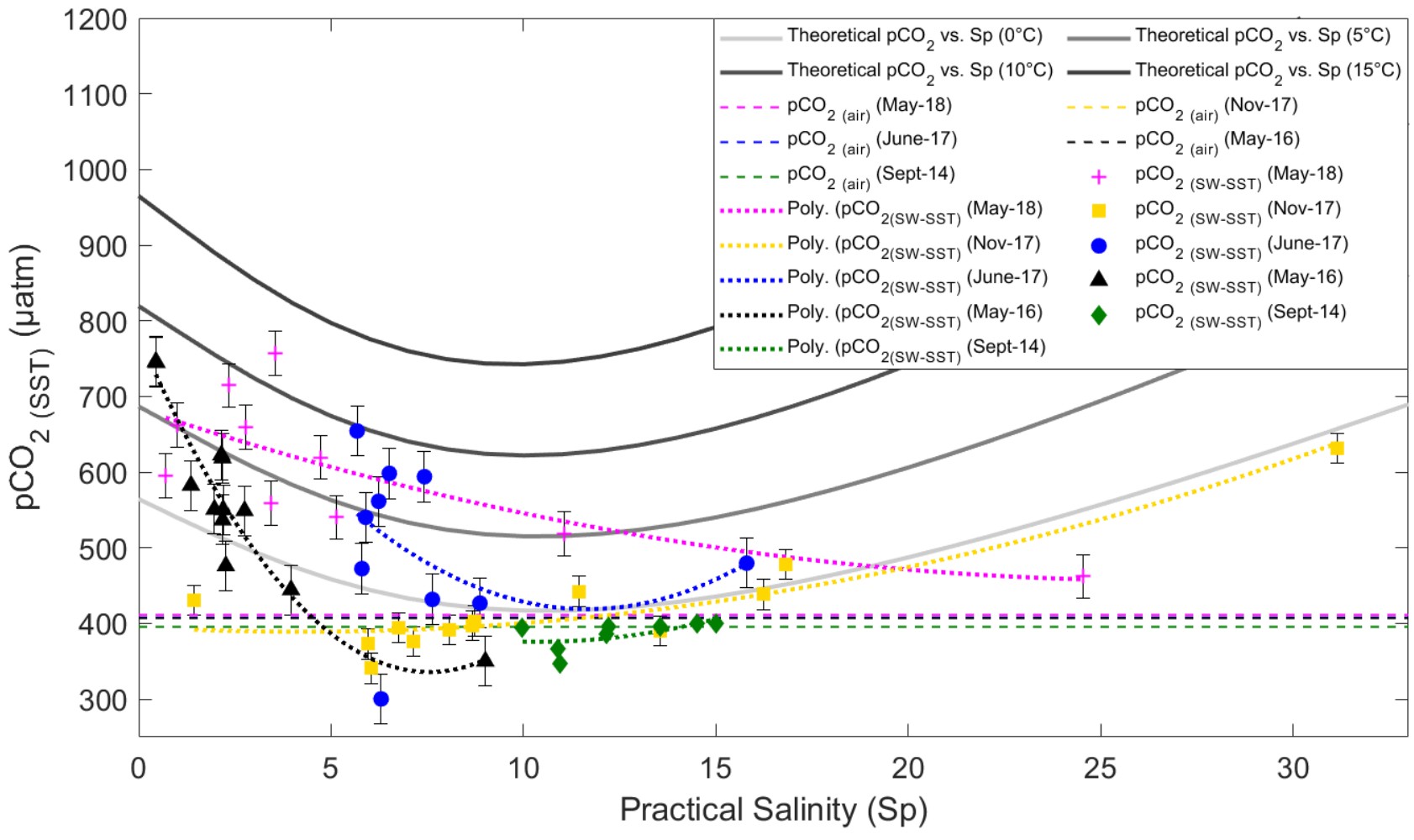

718

**Figure 8. Temperature-normalized field pCO₂(SW-SST), and results of the conservative, two end-member mixing model for pCO₂(SW-SST) in the Saguenay Fjord surface waters. pCO₂(SW-calc) were normalized at each station to the average surface water temperature per sampling month (i.e., T = 10.4°C for September 2014, T = 5.04°C for May 2016, T = 11.9°C for June 2017, T = 7.13°C for November 2017 and T = 5.08°C for May 2018) to account for the effects of temperature on the CO₂ solubility in water, following the procedure described in Jiang et al. (2008). Dashed lines represent the pCO₂(air) on the sampling months (396 ppm in September 2014, 407 ppm in May 2016, 408 ppm in June and November 2017, and 411 ppm in May 2018). Error bars show standard error of the mean for pCO₂(SW-SST) values – bars are smaller than the symbol for September 2014.**

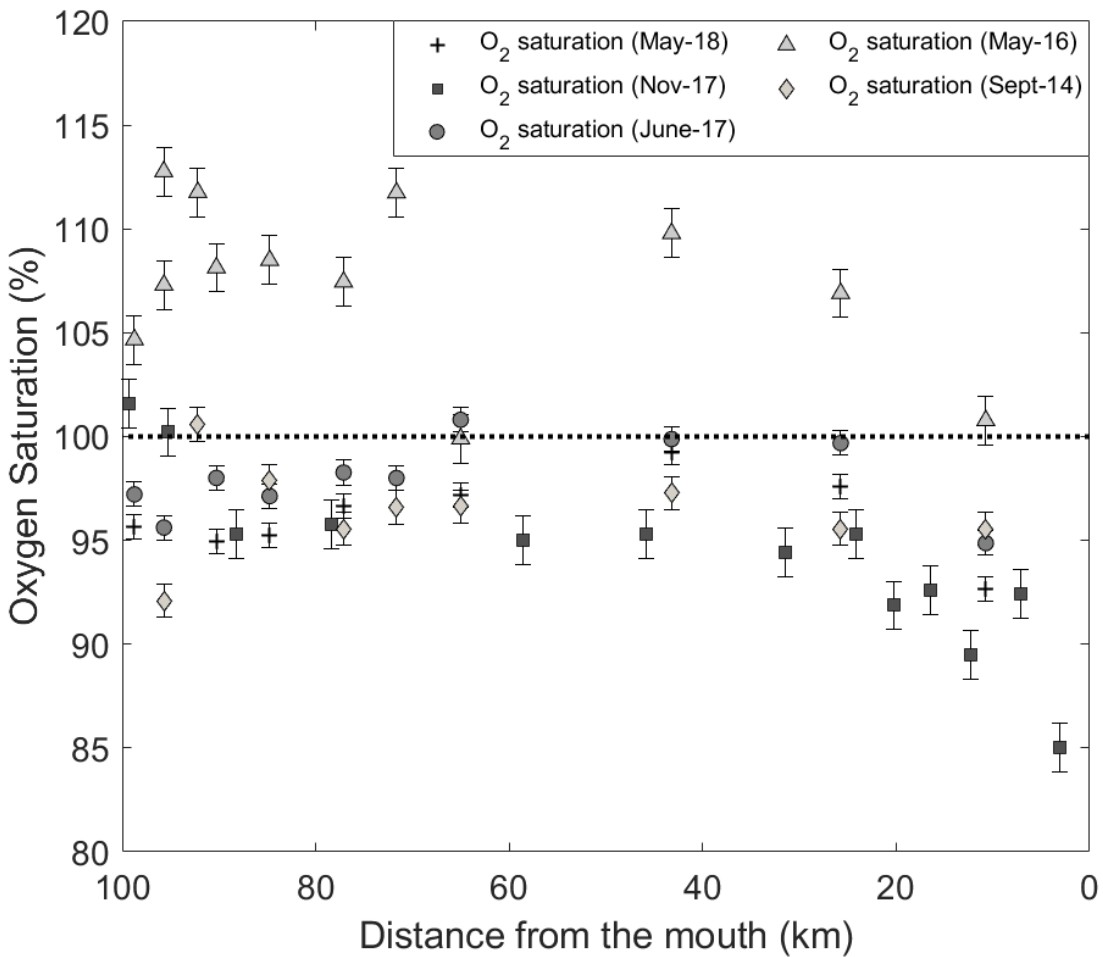

**Figure 9. Spatial distribution of surface-water dissolved O₂ saturation (%) in May 2018, June 2017, November 2017, May**
**2016 and September 2014. The dashed line represents equilibrium with the atmosphere (i.e., 100% saturation). Data points**
**above the line indicate that waters are supersaturated in O₂, whereas those below the line identify O₂-undersaturated waters**
**with respect to the atmosphere. Error bars show standard error of the mean for O₂ saturation values.**

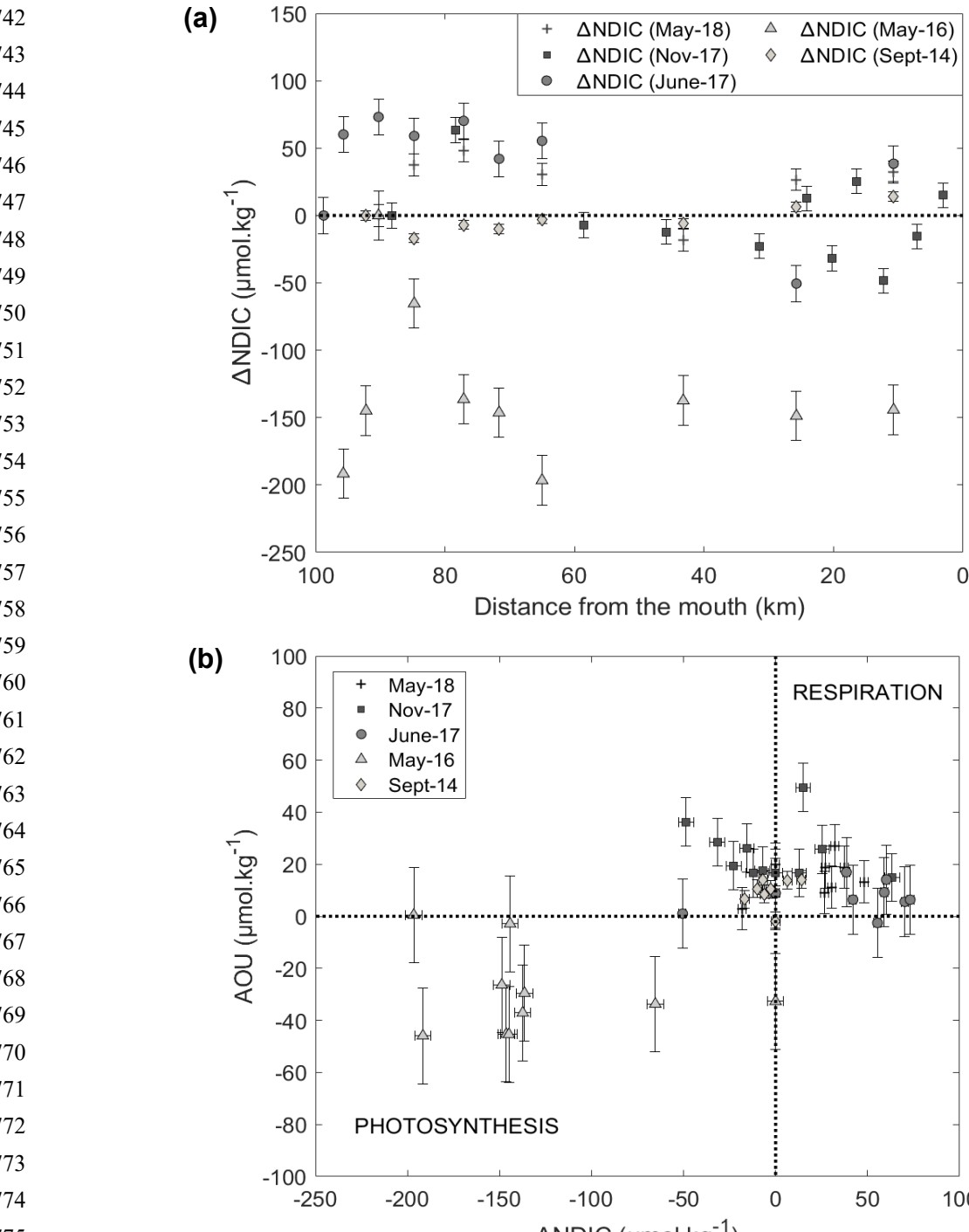

**Figure 10. a)** ΔNDIC (i.e. change in NDIC relative to the saline waters at the head of the fjord) distribution in the Saguenay Fjord surface waters. Data were normalized to a common salinity (average for each sampling period) according to the method of Friis *et al.* **(2003)**; Error bars show standard error of the mean for NDIC values. **b)** Apparent oxygen utilization (AOU) against ΔNDIC. Error bars show standard error of the mean for AOU values.

783  **Tables**

784

| SWT | Salinity | Temperature (°C) | $TA_{(meas)}$ ($\mu mol \cdot kg^{-1}$) | $\delta^{18}O$ (per mil) | DIC ($\mu mol \cdot kg^{-1}$) | DO ($\mu mol \cdot L^{-1}$) |
|---|---|---|---|---|---|---|
| SRW | 0.00 ± 0 | 6.19 ± 0.18 | 154 ± 13 | -12.17 ± 0.21 | 230 ± 12 | 411 ± 6 |
| CIL | 32.52 ± 0.05 | 1.44 ± 0.08 | 2210 ± 2 | -1.12 ± 0.03 | 2141 ± 3 | 256 ± 5 |
| LSLE | 34.31 ± 0.01 | 5.16 ± 0.18 | 2294 ± 2 | -0.17 ± 0.02 | 2276 ± 3 | 76 ± 1 |
| SLRW | 0.00 ± 0 | 12.11 ± 0.13 | 1099 ± 16 | -8.09 ± 0.13 | 1140 ± 15 | 329 ± 5 |
| Weights | 25 | 1 | 25 | 25 | 15 | 1 |

**Table 1. Source-Water Type (SWT) definitions for the Saguenay River (SWR), the St. Lawrence Estuary summertime Cold**
**Intermediate Layer (CIL), the Lower St. Lawrence Estuary bottom water (LSLE) and the St. Lawrence River (SLRW).**
**Definitions and variances were derived from data taken in September 2014, May 2016, June 2017 and November 2017.**
**Data for SRW and SLRW were extrapolated to $S_P = 0$. The weights used in the OMP analysis are also shown.**

| Sampling Month | $pCO_{2(SW-calc)}$ ($\mu atm$) | k ($cm\ h^{-1}$) | u ($m\ s^{-1}$) | F ($mmol \cdot m^{-2} \cdot d^{-1}$) |
|---|---|---|---|---|
| **May 2018** | 623 ± 26 (511/740) | 1.94 ± 0.01 (1.89/1.97) | 3.91 | 6.2 ± 0.79 (2.9/10.0) |
| **November 2017** | 418 ± 12 (353/530) | 3.2 ± 0.04 (2.82/3.38) | 4.2 | 0.40 ± 0.51 (-2.4/4.8) |
| **June 2017** | 506 ± 35 (315/663) | 0.37 ± 0.01 (0.36/0.42) | 1.89 | 0.42 ± 0.15 (-0.4/1.1) |
| **May 2016** | 563 ± 31 (349/724) | 1.26 ± 0.01 (1.15/1.30) | 3.17 | 3.04 ± 0.62 (-1.1/6.5) |
| **September 2014** | 406 ± 6 (369/432) | 1.43 ± 0.01 (1.39/1.49) | 3.71 | 0.16 ± 0.10 (-0.43/0.56) |

792

793  **Table 2. Mean, standard error of the mean and range of $pCO_{2(SW)}$, k, u and F in the Saguenay Fjord surface waters.**
794  **Numbers in parentheses indicate the observed or calculated ranges. Overall, the total area-averaged degassing flux of the**
795  **fjord adds up to 2.14 ± 0.43 $mmol \cdot m^{-2} \cdot d^{-1}$ or 0.78 ± 0.16 $mol \cdot m^{-2} \cdot yr^{-1}$.**

**Data availability**

Data presented in this paper are available upon request from one of the authors (louise.delaigue@mail.mcgill.ca).

**Appendix**

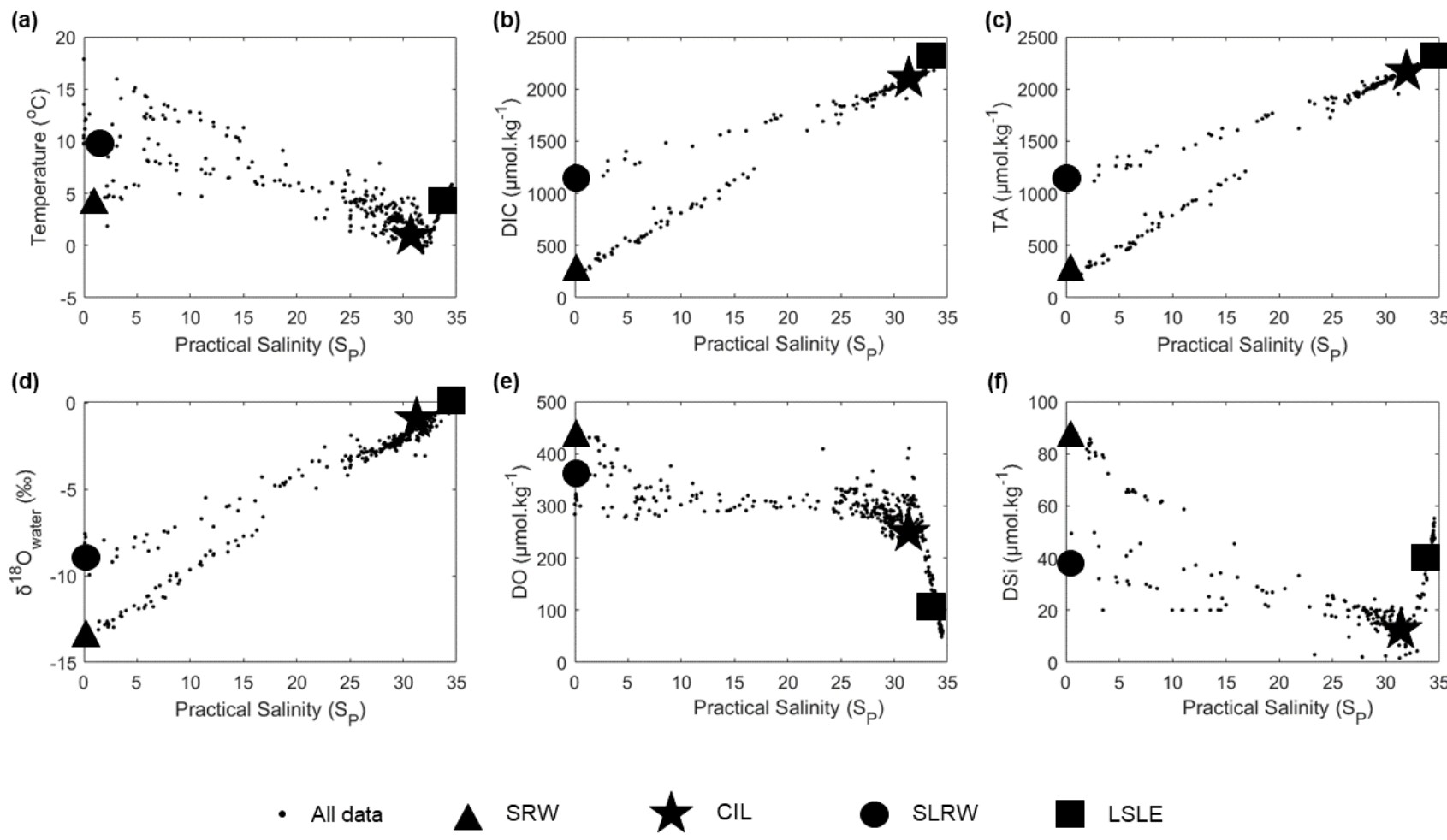

**Figure A. a) Temperature, b) Dissolved Inorganic Carbon (DIC), c) Total Alkalinity (TA), d) $\delta^{18}O_{water}$, e) Dissolved Oxygen (DO), and f) Dissolved silicate (DSi) versus practical salinity (SP) for samples collected in September 2014, May 2016, June 2017, November 2017 and May 2018. Each large symbol (square, circle, triangle and star) identifies distinct source-water masses.**

| Stations / Sampling months | Distance from the mouth (km) | Sept-14 | | May-16 | | June-17 | | May-18 | |
|---|---|---|---|---|---|---|---|---|---|
| | | pH$_{NBS}$ | pH$_T$ | pH$_{NBS}$ | pH$_T$ | pH$_{NBS}$ | pH$_T$ | pH$_{NBS}$ | pH$_T$ |
| St-Fulgence | 98.7 | - | - | 7.348 | - | 7.782 | - | 7.171 | |
| SAG-02 (BHA) | 95.7 | - | - | 7.429 | - | - | - | - | |
| SAG-05 | 95.7 | - | 7.598 | 7.126 | - | 7.478 | - | 7.219 | |
| SAG-06 | 90.3 | - | - | 7.118 | - | 7.559 | - | - | 7.201 |
| SAG-09 (BHA) | 92.2 | - | 7.733 | 7.273 | - | - | - | - | - |
| SAG-15 | 84.8 | - | 7.682 | 7.194 | - | - | 7.413 | - | 7.187 |
| SAG-20 | 77.1 | - | 7.753 | 7.288 | - | - | 7.484 | - | 7.111 |
| SAG-25 | 71.7 | - | 7.772 | 7.348 | - | - | 7.440 | - | - |
| SAG-30 | 65.0 | - | 7.801 | 7.439 | - | - | 7.703 | - | 7.309 |
| SAG-36 | 43.2 | - | 7.758 | 7.351 | - | - | 7.574 | - | 7.359 |
| SAG-42 | 25.7 | - | 7.780 | 7.483 | 7.330 | - | 7.714 | - | 7.394 |
| SAG-48 | 10.7 | - | 7.805 | 7.863 | 7.690 | - | 7.819 | - | 7.604 |

**Figure B.1 Raw in-situ pH data for surface waters collected in September 2014, May 2016, June 2017 and May 2018. Where possible, pH is reported on the total scale, except for freshwater samples which are reported on the NBS scale. No pH data is available on the NBS scale for September 2014 as $10.0 < S_P < 12.4$. Few pH data are available on the total scale for May 2016 as $0.45 < S_P < 9.0$.**

| Stations | Distance from the mouth (km) | Nov-17 | |
| --- | --- | --- | --- |
| | | pH$_{NBS}$ | pH$_T$ |
| SAG-01 | 3.12 | - | 7.862 |
| SAG-02 | 7.00 | - | 7.784 |
| SAG-03 | 12.3 | - | 7.777 |
| SAG-04 | 16.5 | - | 7.755 |
| SAG-05 | 20.3 | - | 7.679 |
| SAG-06 | 24.2 | - | 7.632 |
| SAG-07 | 31.5 | - | 7.637 |
| SAG-08 | 45.9 | - | 7.616 |
| SAG-09 | 58.6 | - | 7.579 |
| SAG-10 | 78.3 | - | 7.589 |
| SAG-11 | 88.3 | - | 7.573 |
| SAG-12 | 95.3 | - | 7.251 |
| SAG-13 | 99.3 | - | 7.608 |

**Figure B.2 Raw in-situ pH data for surface waters collected in November 2017. pH was only reported on the total scale during this cruise, where 1.4 < S$_P$ < 31.2.**

**Author contribution**

A.M. and L.D. conceived the project. A.M. acquired and processed the data prior to 2016. L.D. conducted the data analysis
and wrote the first draft of the paper whereas A.M. provided editorial and scientific recommendations. H.T. provided results
of alkalinity and dissolved inorganic carbon analyses and scientific recommendations.

**Competing interests**

The authors declare that they have no conflict of interest.

**Acknowledgements**

We would like to give special thanks to Gilles Desmeules as well as the Captains and crew of the R/V Coriolis II
without whom, over the years, this project would not have been possible. Most of the data presented in this study were acquired
opportunistically on research cruises funded by Ship-Time Program grants to A.M. or Canadian colleagues by the Natural
Sciences and Engineering Research Council of Canada (NSERC). The work was funded by a Regroupement Stratégique grant
from the Fonds Québécois de Recherche Nature et Technologies (FQRNT) to GEOTOP as well as NSERC Discovery and
505 Marine Environmental Observation, Prediction and Response Network (MEOPAR; Canadian Ocean Acidification Research
partnership) grants to A.M and H.T. We would like to thank Dr. Jean-Francois Hélie at GEOTOP-UQAM for carrying out the
$\delta^{18}O_{water}$ analyses as well as Constance Guignard for cruise preparation and support in the laboratory. Finally, L.D. wishes
to thank MEOPAR and the Department of Earth and Planetary Sciences at McGill for financial support in the form of stipends,
scholarships and assistantships. Bathymetric data of the fjord were graciously provided by Mélanie Belzile. Figure (3) in this
study was created with the Ocean Data View Software (Schlitzer, 2002).

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
