# Peer review of "Spatial variations of CO2 fluxes in the Saguenay Fjord (Québec, Canada) and results of a water mixing model"

_Biogeosciences, 2019_

## Referee Comment (RC1) · Anonymous Referee #1 · 3 Sep 2019

This paper uses an extensive dataset of physical and biogeochemical observations to identify water source contributions to a unique Canadian fjord-type system and evaluate the results in relation to the fjord's air-sea CO2 flux characteristics. Overall, I found this manuscript very well-written, with good explanations of methods used (with one exception I will discuss below), excellent descriptions of data analyses, and clear presentation of results. Also, the paper is concise! While this is very welcome overall, the Introduction and Summary may actually benefit from some additional content.

-The Introduction is a little light. Can more detail be added on coastal CO2 emissions? While there may be little information on CO2 emissions from fjord-like systems, there

have certainly been studies on a variety of other coastal system types which might provide context for this study.

-Similarly, the Summary and Conclusions section is pretty brief. At the very least, what do the authors see as the impacts of this work beyond the studied system. What future work might stem form these findings?

-One very interesting finding of the paper is the negative Org-Alk of the Saguenay River and the fjord waters (Figure 2, manuscript lines 383-388). I am familiar with work detailing positive Org-Alk findings (i.e. calculated TA lower than that measured directly), but I can't think of another example of negative Org-Alk. Negative total alkalinity is common in very acidic waters, but the total alkalinity in this river is positive (although low). This implies to me that in the total alkalinity titration, there is some excess of acid that is not reflected in the pH and DIC measurements. What could this be? This leads me to wish there were more description of the TA measurement method. At which pH range was the titration carried out? What is a shallow end-point detection algorithm? Where might the excess acidity be coming from? A short discussion of the factors that could explain the negative Org-Alk would be a welcome addition.

-The air-sea CO2 flux calculations were based on discrete measurements of DIC and pH at individual stations. However, to produce the overall fluxes for the system, the estuary must have been divided up spatially into segments, as implied by equation 5. However, these segments are not discussed or shown on the map (Figure 1), and should probably be included and delineated in the map.

-Also, pH data were important to this study, but are never shown. At the least it seems that the pH data should be shown in the Appendix figure, but really there should be a discussion of the pH findings before they are used to calculate pCO2.

-In Figure 2, the SRW and CIL TA data are plotted against salinity. It's unclear to me where exactly these data were collected, or how they were selected. The SRW data fall into the salinity range of 0 to ~18 while the CIL data are saltier, from salinity ~22-35.

[Figure]

A regression line is included (although I am skeptical of the R2 of 1.0 shown, given that there is at least some scatter in the data). However, to my eye it seems that the regression line of just the CIL data would produce a different (shallower) slope and (higher) y-intercept that that of the combined SRW and CIL data. If the CIL endmember TA:salinity regression were different, how would that affect the water mass mixing results?

Specific Observations: -L13-L15: this sentence is pretty awkward, can it be simplified? -L26: is there a newer citation for atmospheric CO2 levels than this 2008 work? -L77: the terms "Tmax" and "Sp" have not been defined -L90: the St. Lawrence River and Estuary frequently appear in this manuscript, but it's unclear where these features begin and end in relation to the Saguenay system. -L99: were samples from the St. Lawrence estuary included in the Appendix plot? There seem to be data in this plot that are quite different than those in Figure 2. If so, the locations of the St. Lawrence stations should be shown in Figure 1, and the difference between data from inside and outside the fjord should be clearer. -L110: what is the distinction between "TA" and "TA/DIC" samples? -L124: what is "Rio Tinto Alcan"? -L275: can the location of the weather station be included on the map? What was the measurement height for the wind speed? -L276-277: this is a really nice, concise description of the Schmidt number -L284: specify water temperature here -L298: is there a way to cite or list the conversion formula from NOAA-NWS? -L336: How was the correction for organic alkalinity performed? -L414-424: this correlation analysis assumes that the sensor pCO2 measurements are totally correct; however, there is a fair amount of uncertainty associated with these sensors. Error bars in both the x- and y-directions would be helpful in Figure 5. -Figure 1: the color scale needs a label ('Salinity' etc) -Figure 6(a): the line is dashed-black in my copy, not red as described in the caption -Figure 8: can the mean temperature used to normalize the data be listed somewhere in this figure, for ease of reference?

---

## Referee Comment (RC2) · Anonymous Referee #2 · 18 Sep 2019

Review Manuscript Delaigue et al. 2019

General Comments

Coastal zones play an important hole on the global carbon cycling; however, carbon budges are not yet properly include in global carbon budgets. This paper presents a novel and integrative approach to estimate the relative contribution of known water-sources to the Saguenay Fjord (Quebec, Canada), using geochemical and isotopic tracers coupled with an optimization multiparameter algorithm (OMP). This method, coupled with conservative end-member mixing model, allowed the analysis of dominant factors controlling the CO2 dynamics in the Fjord. The paper is generally well-written

and very easy to follow, providing new insights on coastal carbon dynamics. The paper is very succinct, and this is welcome. However, in some passages I would like to see more advances beyond the studied area. In brief, the manuscript lacks to present a better contextualization and to describe the implications of these findings. But of course this does diminish the merits of this manuscript.

The introduction is too short. In recent years, the knowledge of CO2 dynamics was considerable increased in coastal zones worldwide. In this way, I strongly recommend a review of the literature to contextualize your research. In addition, the discussion section is also very short, especially when discussing the governing processes that drive the concentrations and fluxes of CO2 at the air-water interface in the estuary.

The methodology is overall well written, however I have some doubts especially about the OMP analysis. How did you weight "arbitrarily" the parameters included in the OMP calculations? Another question: you argued, "Each source-water type is only appropriate for the fjord and for the period of study". The source-water type definitions were the Saguenay River (SWR), the St. Lawrence Estuary summertime Cold Intermediate Layer (CIL), the Lower St. Lawrence Estuary bottom waters (LSLE) and the St. Lawrence River (SLRW). The sampling campaigns were performed in late spring (May 2016 and May 2018), early summer (June 2017), and early and late fall (September 2014 and November 2017). I mean, the considered water masses encompass all characteristics of the sampled periods? Are there significant differences in the end-members considering these different seasons? Looking at the Appendix, there are some scattering in the end-members of SRW, CIL, SLRW and LSLE. Could this cause influences when calculating the OMP and the mixing model end-members?

The discussion of negative organic alkalinity should be better stressed in the manuscript. This is a very atypical pattern, taking into account that almost all studies that investigate organic alkalinity in coastal zones found positive concentrations. Another point: How did you correct the values of TA (organic alkalinity) to compute the mixing models?

Specific Comments

Line 26 : As you are talking about the concentrations of CO2 in the past, I recommend to include the study of Willeit et al (2019), which suggests that "the current CO2 concentration is unprecedented over the past 3 million years".

Willeit1, M., Ganopolski, A., Calov, R., Brovkin, V. Mid-Pleistocene transition in glacial cycles explained by declining CO2 and regolith removal. Science Advances, Vol. 5, no. 4, eaav7337. DOI: 10.1126/sciadv.aav7337

Line 28: Here, I think the good reference is Feely et al. (2004).

Feely, R. A. 2004. Impact of Anthropogenic CO2 on the CaCO3 System in the Oceans. Science 305, 362.

Line 31: I could not find this reference. Is it Caldeira and Wickett (2005)?

Line 38-40: This sentence is not clear.

Line 49: What do you refers to trophic status? According to Vollenweider et al. (1998), trophic conditions of marine waters are related to degree of nutrient enrichment. Oligotrophy means nutrient poor (low productivity) and eutrophy means nutrient rich (high productivity) waters. However, the analysis of trophic status "per se" do not give information whether the ecosystems is a source or a sink of CO2 to the atmosphere.

Vollenweider, R. A., Giovanardi, F., Montanari, G., Rinaldi, A. 1998. Characterization of the trophic conditions of marine coastal waters with special reference to the NW Adriatic Sea: proposal for a trophic scale, turbidity and generalized water quality index. Environmetrics, 9, 329-357.

Line 61: I could not find these tributaries in the Fig. 1b.

Lines 80-81: Please, give the range of temperature for the warm brackish surface layer of the St. Lawrence Estuary. What is the tidal amplitude in the Fjord, and the longitudinal variations? Could you include this information?

Lines 132-142: Why did you use different methodologies of pH measurements for Sp >5 (spectrophotometry) and Sp < 5 (potentiometric)? Did you investigate the differences between these methods?

Lines 148-149: It no was clear how you did convert the pHNBS to pHT. Could you explain this procedure in the manuscript? Did you apply correction factors for the pH measurements at NBS scale for the TRIS buffer solutions (for which you have assigned the pHT)?

Line 158: What is the concentration of CO2 that you insert in the vials?

Line 189: "…biogeochemical cycling is imperative if one is to evaluate the movement of nutrients…". Something is missing here.

Lines 214-222: This passage is somewhat confuse. I think you should explain about this "arbitrary choices" in the weighting procedure based on covariance between tracers.

Lines 226-225: "In the context of biogeochemical cycles, a SWT should be defined where the water mass enters the basin, upstream from the mixing region (Karstensen, 2013)." However, if the water masses enter the basins downstream from the mixing region?

Lines 229-233: You argued that "Each definition was captured relative to the fjord, i.e. each source-water type is only appropriate for the fjord and for the period of study". Are you sure that these chosen SWT are representative for the period of study (late spring, May 2016 and May 2018; early summer, June 2017; early and late fall, September 2014 and November 2017)? In addition, did you take into account the seasonal variability of the end-members to calculate the OMP and the mixing models?

Line 265: ðİŘź = −ðİŘů ðİŻ£ðİŚŘ/ðİŻ£ðİŚě. Provide the terms of the equation.

Line 270: The parameterization of Wanninkhof (2014) is recommended for calculations of air-water exchanges in open ocean waters. I think you should include here other

parameterization more appropriate for estuarine environments.

Line 305: It no is clear to me how you separated these segments for the fjord's surface area. Did you separate by salinity? Distance from the mouth?

Lines 383-385: The discussion of the negative organic alkalinity results are poorly presented. I recommend put more efforts in this subject.

Lines 414-420: You attributed the average difference between pCO2(SW-meas) and pCO2(SW-calc) to the uncertain associated with the carbonic acid dissociation constants. One possible alternative is to calculate the pCO2(SW-calc) using other available constants to investigate which one fits better with the pCO2(SW-meas).

Lines 435-446: This paragraph is very interesting, but I missed the comparison with other studies that applied end-member mixing models, contrasting the influences of mixing and biological activities.

Lines 447-457: Where are the results of the fluorometer? I think this section can be strengthened adding with these results. For example you agueed that "Additionally, it is interesting to note that NDIC is chronically negative for all sampling months near the 45 km mark." Maybe the fluorescence call tell something.

Fig. 1b. Please, provide the title of the Y-right axis. In addition, add the riverine positions in the figure and the estuarine sections you used to calculate the air-water CO2 fluxes.

Fig. 10. Normally, the comparison of DIC and AOU are performed by calculating the excess of dissolved organic carbon (E-DIC), which is difference between the in situ DIC and a theoretical DIC at atmospheric equilibrium. Are there differences comparing ∆NDIC x AOU with E-DIC x AOU?

[Figure]

---

## Author Comment (AC1) · 29 Oct 2019

**Responses to Anonymous Referee #1**
**(Responses to the referee's comments are in bold)**

This paper uses an extensive dataset of physical and biogeochemical observations to identify water source contributions to a unique Canadian fjord-type system and evaluate the results in relation to the fjord's air-sea $CO_2$ flux characteristics. Overall, I found this manuscript very well-written, with good explanations of methods used (with one exception I will discuss below), excellent descriptions of data analyses, and clear presentation of results. Also, the paper is concise!

**We thank the referee for his(her) detailed and very positive comments.**

While this is very welcome overall, the Introduction and Summary may actually benefit from some additional content. -The Introduction is a little light. Can more detail be added on coastal $CO_2$ emissions? While there may be little information on $CO_2$ emissions from fjord-like systems, there have certainly been studies on a variety of other coastal system types which might provide context for this study.

**A few sentences will be added to the revised manuscript to summarize the current consensus about $CO_2$ emissions in estuarine and coastal environments, and the introduction will read:**

[revised manuscript text omitted]

Rhein, M., S.R. Rintoul, S. Aoki, E. Campos, D. Chambers, R.A. Feely, S. Gulev, G.C. Johnson, S.A. Josey, A. Kostianoy, C. Mauritzen, D. Roemmich, L.D. Talley and F. Wang (2013): Observations: Ocean. In: Climate Change 2013: The Physical Science Basis. Contribution of Working Group I to the Fifth Assessment Report of the Intergovernmental Panel on Climate Change [Stocker, T.F., D. Qin, G.-K. Plattner, M. Tignor, S.K. Allen, J. Boschung, A. Nauels, Y. Xia, V. Bex and P.M. Midgley (eds.)]. Cambridge University Press, Cambridge, United Kingdom and New York, NY, USA.

Rysgaard, S., Mortensen, J., Juul-Pedersen, T., Sørensen, L. L., Lennert, K., Søgaard, D. H., … Bendtsen, J. (2012) High air–sea $CO_2$ uptake rates in nearshore and shelf areas of Southern Greenland: Temporal and spatial variability. Marine Chemistry, 128–129, 26–33. https://doi.org/10.1016/j.marchem.2011.11.002

Sabine, C. L, (2004) The oceanic sink for anthropogenic $CO_2$. Science. 305(5682), 367–371. doi:10.1126/science.1097403.

Smith, R. W., Bianchi, T. S., Allison, M., Savage, C., and Galy, V. (2015) High rates of organic carbon burial in fjord sediments globally. Nature Geoscience, 8(6), 450–453. https://doi.org/10.1038/ngeo2421

Willeit, M., Ganopolski, A., Calov, R., & Brovkin, V. (2019) Mid-Pleistocene transition in glacial cycles explained by declining $CO_2$ and regolith removal. Science Advances, 5(4), eaav7337.

-Similarly, the Summary and Conclusions section is pretty brief. At the very least, what do the authors see as the impacts of this work beyond the studied system? What future work might stem from these findings? –

A few sentences will be added to the revised manuscript to summarize the impacts of this work beyond the studied system, extend the conclusions of the data analysis to an understanding of the factors governing $CO_2$ fluxes at the air-water interface and how these might apply to other systems (regardless of scale).

"Studying the carbon budget of fjord inlets not only provides information on its trophic status (i.e. source or sink of $CO_2$ with respect to the atmosphere) and surface-water chemistry, but also explores the magnitude of gas exchange and the amount of biological activity it sustains. In addition to biological production, upwelling, water temperature, and the spreading of freshwater plumes all regulate $pCO_2$ in costal systems. Wind speed is also critical in estimating gas exchange at the air-sea interface as it heavily impacts sea state (Chen et al., 2013). The importance of wind on controlling the $CO_2$ flux needs to be further investigated, especially at high latitudes where strong winds are often encountered (Chen et al., 2013) and in narrow inlets where the fetch is limited. Anthropogenic activities are altering the continental water cycle, along with the flows of carbon, nutrients and sediment to the coastal oceans (Borges, 2005), and hence, the sequestration of anthropogenic $CO_2$ emissions by the oceans. Current research on $CO_2$ fluxes in coastal zones is still too scarce to make precise climate change predictions (i.e., flux within $\pm 0.05$ Pg·C $y^{-1}$) on whether they mitigate or accelerate atmospheric $CO_2$ emissions."

One very interesting finding of the paper is the negative Org-Alk of the Saguenay River and the fjord waters (Figure 2, manuscript lines 383-388). I am familiar with work detailing positive Org-Alk findings (i.e. calculated TA lower than that measured directly), but I can't think of another example of negative Org-Alk. Negative total alkalinity is common in very acidic waters, but the total alkalinity in this river is positive (although low). This implies to me that in the total alkalinity titration, there is some excess of acid that is not reflected in the pH and DIC measurements. What could this be? This leads me to wish there were more description of the TA measurement method. At which pH range was the titration carried out? What is a shallow end-point detection algorithm? Where might the excess acidity be

coming from? A short discussion of the factors that could explain the negative Org-Alk would be a welcome addition.

**The negative Org-Alk (acidity) component of this manuscript could make up a manuscript of its own and a detailed discussion is, therefore, beyond the scope of this study. For a discussion of the acid-base properties of dissolved organic matter in estuaries and the impact of OrgAlk on pH, acid-base dissociation and carbonic acid speciation, the reviewer should consult Cai et al. (1998) and Muller and Bleie (2008). Although positive values are most often reported, negative organic alkalinities (acidity) have been reported in coastal waters (e.g., Yang et al., 2015) and discussed (Ulfsbo et al., 2015). They are relatively common in rivers and stream waters of temperate regions where soil profiles are well developed and the bedrock is made up of crystalline rocks (igneous or metamorphic silicates) devoid of carbonates. In fact, all the rivers along the north shore of the St. Lawrence Estuary are characterized by circum-neutral pHs and negative Org-Alk (acidity) as they drain the metamorphic/igneous rocks of the Canadian Shield (Wilkinson et al., 1992). The negative Org-Alk (acidity) most likely originates from soil humic acids and all these rivers, including the Saguenay River, are highly colored. We added the following text to the revised manuscript:**

**"The negative Org-Alk (acidity) of the Saguenay River water most likely originates from soil humic acids that are flushed by percolation with groundwaters that drain the metamorphic and igneous rocks of the Canadian Shield."**

**Total alkalinity was measured by metered weak acid solution additions between pH 7 and 4, sometimes requiring flushing of the sample solution with $N_2$ to lower the $pCO_2$ and increase the initial pH. The shallow end-point detection algorithm is a proprietary software of the manufacturer, Radiometer, for weak acid/base potentiometric titration.**

**Cai W. J., Wang Y. and Hodson R. E. (1998) Acid-base properties of dissolved organic matter in the estuarine waters of Georgia, USA. Geochim. Cosmochim. Acta 62, 761 473–483**

**Muller F. L. L. and Bleie B. (2008) Estimating the organic acid contribution to coastal seawater alkalinity by potentiometric titrations in a closed cell. Anal. Chim. Acta 619, 183–191.**

**Ulfsbo A., Kuliński K., Anderson L. G. and Turner D. R. (2015) Modelling organic alkalinity in the Baltic Sea using a Humic-Pitzer approach. Mar. Chem. 168, 18–26.**

**Wilkinson K.J., Jones H.G., Campbell P.G.C. and Lachance M. (1992) Estimating organic acid contributions to surface water acidity in Quebec (Canada). Water, Air and Soil Pollution 61, 57-74.**

**Yang B., Byrne R.H. and Lindemuth M. (2015) Contributions of organic alkalinity to total alkalinity in coastal waters: A spectrophotometric approach. Mar. Chem. 176, 199-207.**

-The air-sea $CO_2$ flux calculations were based on discrete measurements of DIC and pH at individual stations. However, to produce the overall fluxes for the system, the estuary must have been divided up spatially into segments, as implied by equation 5. However, these segments are not discussed or shown on the map (Figure 1), and should probably be included and delineated in the map.

**The fjord was divided in segments based on the overall trend of the surface water $pCO_2$ ($pCO_{2(sw)}$) along the main axis of the fjord (Fig. 4): the first segment includes the larger inner basin (over which**

pCO$_{2(SW)}$ is much higher than pCO$_{2(air)}$ and decreases rapidly downstream) whereas the second segment encompasses the two outer basins (over which pCO$_{2(SW)}$ is close to pCO$_{2(air)}$ and varies little downstream). Segments will be identified on Figure 1.a.

-Also, pH data were important to this study, but are never shown. At the least it seems that the pH data should be shown in the Appendix figure, but really there should be a discussion of the pH findings before they are used to calculate pCO$_2$.

**A table will be added to the Appendix with the raw pH data.**

-In Figure 2, the SRW and CIL TA data are plotted against salinity. It's unclear to me where exactly these data were collected, or how they were selected. The SRW data fall into the salinity range of 0 to ~18 while the CIL data are saltier, from salinity ~22-35. A regression line is included (although I am skeptical of the R2 of 1.0 shown, given that there is at least some scatter in the data). However, to my eye it seems that the regression line of just the CIL data would produce a different (shallower) slope and (higher) y-intercept that that of the combined SRW and CIL data. If the CIL endmember TA:salinity regression were different, how would that affect the water mass mixing results?

**$R^2$ = 0.999, which was rounded to 1 for ease of reference. It will be modified back to 0.999.**

**The reviewer is right (see below) in saying that the slope of the regression line to the CIL data is slightly shallower (TA/S$_P$ = 47.8 vs 63.4) but the extrapolations to S$_P$ = 0 (the SRW endmember) from the low salinity data (S$_P$ < 11) alone are nearly identical (TA = 153 vs 147 µmol/kg). Hence, the water mass mixing results would not be affected significantly if the data were binned. The CIL definition was taken directly at the source of the endmember. As few data (0 < S$_P$ < 5) were available to define the SRW, we used the full set of data from the fjord water column to correlate TA and S$_P$ and extrapolate the definition of the SRW endmember to S$_P$ = 0.**

[Figure]

[Figure]

[Figure]

-L13-L15: this sentence is pretty awkward, can it be simplified?

**The sentence will be simplified to ease understanding. In the revised manuscript, it will read:**

**"Nonetheless, the $CO_2$ dynamics in the fjord are modulated with the rising tide by the intrusion, at the surface, of brackish water from the upper estuary, as well as an overflow of mixed seawater over the shallow sill from the lower estuary."**

-L26: is there a newer citation for atmospheric $CO_2$ levels than this 2008 work?

**The other referee suggested Willeit et al (2019), which will be incorporated in the revised manuscript**

- L77: the terms "Tmax" and "Sp" have not been defined

**The terms will be defined in the revised manuscript. "Tmax" will be changed to "T", which stands for Temperature (in °C) whereas "S_P" refers to the practical salinity of the waters.**

-L90: the St. Lawrence River and Estuary frequently appear in this manuscript, but it's unclear where these features begin and end in relation to the Saguenay system.

**The original inset in Figure 1 will be replaced by the following map that also includes the location of sampling stations in the estuary and the gulf from which the SLR, CIL and LSLE endmember definitions were derived.**

[Figure]

-L99: were samples from the St. Lawrence estuary included in the Appendix plot? There seem to be data in this plot that are quite different than those in Figure 2. If so, the locations of the St. Lawrence stations should be shown in Figure 1, and the difference between data from inside and outside the fjord should be clearer.

**Samples from the St. Lawrence Estuary were included in the Appendix plot, specifically stations used to define endmembers other than the SRW. The location of sampling sites outside the fjord for which data are used in this plot will be identified in the new inset map (see above) and are identified by letters (A to K) in the Upper Estuary and numbers (18 to 25) in the Lower Estuary and the Gulf. The data taken inside and outside the fjord are distinguished by distinct symbols for each water mass.**

-L110: what is the distinction between "TA" and "TA/DIC" samples?

**"TA" refers to samples collected in 250 mL glass bottles throughout the water column and analyzed at McGill University. "TA/DIC" refers to surface water samples taken in 500 mL glass bottles and sent to Dalhousie University to be analysed by Dr. Helmuth Thomas for both TA and DIC. Methods, Lines 170-184 describe the analytical methods in detail.**

-L124: what is "Rio Tinto Alcan"?

**Rio Tinto Alcan is a large multinational aluminum smelter/producer. The company constructed and manages its own hydroelectric dam on the Saguenay River. We collaborated with a Water Management Consultant who provided us with freshwater discharge data as part of their bank stabilization programme.**

-L275: can the location of the weather station be included on the map? What was the measurement height for the wind speed?

**The weather station location will be included on the map. The weather station's elevation is 152 m above sea level.**

-L276-277: this is a really nice, concise description of the Schmidt number

**We thank the referee for his(her) positive comment**

-L284: specify water temperature here

**L284 will be rephrased in the revised manuscript and will read: "[…] where T is the temperature (°C) and A, B, C, D and E are fitting coefficients for seawater ($S_P$ = 35) and freshwater ($S_P$ =0), for water temperatures ranging from -2°C to 40°C (Wanninkhof, 2014)."**

-L298: is there a way to cite or list the conversion formula from NOAA-NWS?

**Unfortunately, we have not found a way of citing Tim Brice's (very useful!) work. However, a portable version of the Weather Calculator is now available (here)**

-L336: How was the correction for organic alkalinity performed?

**Line 320: "*The organic alkalinity of the fjord waters was estimated from the difference between the measured and calculated TA*". The organic alkalinity was then subtracted from TA$_{meas}$ to give TA$_{calc}$.**

-L414-424: this correlation analysis assumes that the sensor $pCO_2$ measurements are totally correct; however, there is a fair amount of uncertainty associated with these sensors. Error bars in both the x- and y-directions would be helpful in Figure 5.

**As noted in the manuscript, "the manufacturer claims a 1% accuracy, but the performance of the instrument may be even better (Hunt et al., 2017)", an insignificant instrumental error. However, surface water $pCO_2$s recorded by the probe can vary by as much as 5% as the ship drifts from its position, water flows past the ship and probe, or waters are mixed by turbulence. Error bars will be added to Figure 5 in the revised manuscript.**

-Figure 1: the color scale needs a label ('Salinity' etc)

**The proper label will be added to Figure 1 in the revised manuscript.**

-Figure 6(a): the line is dashed-black in my copy, not red as described in the caption

**We thank the referee for catching this mistake! The figure caption will be modified accordingly in the revised manuscript.**

-Figure 8: can the mean temperature used to normalize the data be listed somewhere in this figure, for ease of reference?

**Temperatures used to normalize the data were listed on lines 430-432 of the original manuscript but will be listed in the figure caption for ease of reference.**

---

## Author Comment (AC2) · 29 Oct 2019

**Responses to Anonymous Referee #2**
**(Responses to the referee's comments are in bold)**

Coastal zones play an important hole on the global carbon cycling; however, carbon budges are not yet properly include in global carbon budgets. This paper presents a novel and integrative approach to estimate the relative contribution of known water sources to the Saguenay Fjord (Quebec, Canada), using geochemical and isotopic tracers coupled with an optimization multiparameter algorithm (OMP). This method, coupled with conservative end-member mixing model, allowed the analysis of dominant factors controlling the $CO_2$ dynamics in the Fjord. The paper is generally well-written and very easy to follow, providing new insights on coastal carbon dynamics. The paper is very succinct, and this is welcome. However, in some passages I would like to see more advances beyond the studied area. In brief, the manuscript lacks to present a better contextualization and to describe the implications of these findings. But of course this does diminish the merits of this manuscript. The introduction is too short. In recent years, the knowledge of $CO_2$ dynamics was considerable increased in coastal zones worldwide. In this way, I strongly recommend a review of the literature to contextualize your research.

**We thank the referee for his(her) detailed and very positive comments.**
**A few sentences will be added to the revised manuscript to summarize the current consensus about $CO_2$ emissions in estuarine and coastal environments.**

**Please see response to Referee #1.**

In addition, the discussion section is also very short, especially when discussing the governing processes that drive the concentrations and fluxes of $CO_2$ at the air-water interface in the estuary.

**A few sentences will be added to the revised manuscript:**

**"These results highlight the importance of the freshwater plume from the Saguenay River in regulating the $pCO_2$ dynamics in the fjord. Winds, in addition to regulate the gas exchange coefficient, are also known to have a direct influence on air-sea $CO_2$ fluxes by driving upwelling of $CO_2$-rich waters along with the entrainment of nutrients in surface waters, thus increasing biological activity (Wanninkhof and Triñanes, 2017). However, wind speeds are relatively low in the studied system ($1.89 \text{ m s}^{-1} < u < 4.2 \text{ m s}^{-1}$, Table 2), implying a calm sea state (Frankignoulle, 1998), and hence reinforcing that changes in $pCO_{2(SW-SST)}$ can mainly be attributed to microbial respiration and photosynthesis modulated by water renewals rather than winds."**

**Frankignoulle, M. (1988) Field measurements of air-sea $CO_2$ exchange 1. Limnology and Oceanography, 33(3), 313-322.**

**Wanninkhof, R., and Triñanes, J. (2017) The impact of changing wind speeds on gas transfer and its effect on global air-sea $CO_2$ fluxes. Global Biogeochemical Cycles, 31(6), 961-974.**

The methodology is overall well written; however I have some doubts especially about the OMP analysis. How did you weight "arbitrarily" the parameters included in the OMP calculations?
*Coupled with specific comment:* Lines 214-222: This passage is somewhat confuse. I think you should explain about this "arbitrary choices" in the weighting procedure based on covariance between tracers.

**Parameters were weighted arbitrarily according to their mixing behaviors (i.e., whether they behave conservatively or not, are affected by biological activity or gas exchange across the air-water interface) following [Lansard, B., Mucci, A., Miller, L. A., Macdonald, R. W., and Gratton, Y.: Seasonal variability of water mass distribution in the southeastern Beaufort Sea determined by total alkalinity and $\delta^{18}O$, J. Geophys. Res-Oceans, 117, 2012.]. Furthermore, several OMP analyses were carried out using different weights for each parameter while always considering their conservative behavior (i.e., low, medium or high) and results were not affected significantly. To clarify, the following text will be added to the revised manuscript:**
**"Several OMP analyses were carried out using different weights for each parameter, while weighing their conservative behaviour appropriately (i.e., highly conservative vs. lightly conservative). Results were not affected significantly."**

Another question: you argued, "Each source-water type is only appropriate for the fjord and for the period of study". The source-water type definitions were the Saguenay River (SWR), the St. Lawrence Estuary summertime Cold Intermediate Layer (CIL), the Lower St. Lawrence Estuary bottom waters (LSLE) and the St. Lawrence River (SLRW). The sampling campaigns were performed in late spring (May 2016 and May 2018), early summer (June 2017), and early and late fall (September 2014 and November 2017). I mean, the considered water masses encompass all characteristics of the sampled periods? Are there significant differences in the end-members considering these different seasons? Looking at the Appendix, there are some scattering in the end-members of SRW, CIL, SLRW and LSLE. Could this cause influences when calculating the OMP and the mixing model end-members?

**A seasonality analysis was carried out in order to make sure the SWT definitions are appropriate for the period of study. Insignificant variations were observed in tracers such as $^{18}O$, DIC, TA, DO and $S_P$. The only significantly variable tracer was T, which was given the lowest possible weight in the OMP analysis as to not skew the water mass analysis results.**

**We are currently writing a manuscript in which we tackle this issue in depth, including seasonal variations of bottom-water renewals in the fjord. It will include a thorough analysis of the seasonality of the SWT definitions.**

The discussion of negative organic alkalinity should be better stressed in the manuscript. This is a very atypical pattern, taking into account that almost all studies that investigate organic alkalinity in coastal zones found positive concentrations.

**As noted in our response to Reviewer#1's inquiry, negative organic alkalinities (acidity) in rivers are relatively common in temperate regions where soil profiles are well developed and the bedrock is made up of crystalline rocks (igneous or metamorphic silicates) devoid of carbonates. In fact, all the rivers along the north shore of the St. Lawrence Estuary are characterized by circum-neutral pHs and negative Org-Alk (acidity) as they drain the metamorphic/igneous rocks of the Canadian Shield. The negative Org-Alk (acidity) most likely originates from soil humic acids and all these rivers, including the Saguenay River, are highly colored.**

Another point: How did you correct the values of TA (organic alkalinity) to compute the mixing models?

**Line 320: "*The organic alkalinity of the fjord waters was estimated from the difference between the measured and calculated TA*". To avoid organic alkalinity skewing the results, TA was calculated ($TA_{calc}$) using DIC and pH. The corrected TA were then used in the mixing model.**

Line 26 : As you are talking about the concentrations of $CO_2$ in the past, I recommend to include the study of Willeit et al (2019), which suggests that "the current $CO_2$ concentration is unprecedented over the past 3 million years".

**We thank the referee for his(her) suggestion, as the other referee also recommended the use of a more recent reference.**

Line 28: Here, I think the good reference is Feely et al. (2004).
Feely, R. A. 2004. Impact of Anthropogenic CO2 on the CaCO3 System in the Oceans. Science 305, 362.

**The reference will be added to the revised manuscript.**

Line 31: I could not find this reference. Is it Caldeira and Wickett (2005)?

**Yes, it is.  The in-text citation will be modified accordingly.**

Line 38-40: This sentence is not clear.

**The sentence will be simplified to ease understanding. It will read:**

**"High latitude waters such as the Arctic Ocean have recently been given most of the attention, while coastal, seasonally ice-covered aquatic environments, such as the Saguenay Fjord, display comparable inter-annual and climatic sea-ice cover variabilities all the while being much more accessible (Bourgault et al., 2012)."**

Line 49: What do you refers to trophic status? According to Vollenweider et al. (1998), trophic conditions of marine waters are related to degree of nutrient enrichment. Oligotrophy means nutrient poor (low productivity) and eutrophy means nutrient rich (high productivity) waters. However, the analysis of trophic status "per se" do not give information whether the ecosystems is a source or a sink of $CO_2$ to the atmosphere.
Vollenweider, R. A., Giovanardi, F., Montanari, G., Rinaldi, A. 1998. Characterization of the trophic conditions of marine coastal waters with special reference to the NW Adriatic Sea: proposal for a trophic scale, turbidity and generalized water quality index. Environmetrics, 9, 329-357.

**Trophic status is indeed directly linked to primary productivity and microbial respiration. In our definition of the trophic status, we differentiate between surface waters that are net sources and net sinks of $CO_2$ to the atmosphere. An autotrophic system will generally be a sink of $CO_2$ to the atmosphere whereas a heterotrophic system will generally be a source, but there might be exceptions in transition zones between $CO_2$-charged waters and productive estuarine waters.**

Line 61: I could not find these tributaries in the Fig. 1b.

**Tributaries will be added to Fig. 1a of the revised manuscript. We thank the referee for catching this mistake.**

Lines 80-81: Please, give the range of temperature for the warm brackish surface layer of the St. Lawrence Estuary.

**The range of temperatures for the warm brackish surface layer of the St. Lawrence Estuary will be added to the revised manuscript.**

What is the tidal amplitude in the Fjord, and the longitudinal variations? Could you include this information?

**The requested information will be added to the revised manuscript.**

**According to Seibert et al. (1979), the tidal amplitude at the mouth of the fjord near Tadoussac averages 4.0 m and increases slightly toward the head of the fjord (4.3 m near Port Alfred). Spring tides may reach an amplitude of 6 m.**

**Seibert, G. H., Trites, R. W., and Reid, S. J. (1979) Deepwater exchange processes in the Saguenay Fjord, J. Fish. Board Can., 36(1), 42– 53.**

Lines 132-142: Why did you use different methodologies of pH measurements for Sp >5 (spectrophotometry) and Sp < 5 (potentiometric)? Did you investigate the differences between these methods?

**The differences between these methods have been investigated by Mucci's research group over many cruises in the St. Lawrence Estuary and the Saguenay Fjord over the past 15 years. Low salinity waters ($S_P$ < 5) are often colored, turbid and poorly buffered and, thus, are often not amenable to spectrophotometric measurements with colored dyes.**

Lines 148-149: It no was clear how you did convert the pHNBS to pHT. Could you explain this procedure in the manuscript? Did you apply correction factors for the pH measurements at NBS scale for the TRIS buffer solutions (for which you have assigned the pHT)?

**We calculated the difference between the assigned $pH_T$ of a TRIS buffer of salinity close to the sample (±2.5) and the measured pH(NBS) of the TRIS buffer. The sample pH(NBS) was then converted to $pH_T$ by subtracting this value from the measured pH(NBS) of the sample.**

Line 158: What is the concentration of $CO_2$ that you insert in the vials?

**99.998% pure $CO_2$ (Research Grade) was injected in dual inlet mode.**

Line 189: ". . .biogeochemical cycling is imperative if one is to evaluate the movement of nutrients. . .". Something is missing here.

**The sentence starts on line 188 and reads as follows: "Resolving the effects of mixing and biogeochemical cycling is imperative if one is to evaluate the transport of nutrients and tracers in a water body."**

Lines 226-225: "In the context of biogeochemical cycles, a SWT should be defined where the water mass enters the basin, upstream from the mixing region (Karstensen, 2013)." However, if the water masses enter the basins downstream from the mixing region?

**By "upstream from the mixing region", we mean before the SWT enters the mixing region, and therefore at the source of the SWT itself. The sentence will be reworded for clarity and will read:**

**"In the context of biogeochemical cycles, a SWT should be defined where the water mass enters the basin, before it enters the mixing region (Karstensen, 2013)."**

Lines 229-233: You argued that "Each definition was captured relative to the fjord, i.e. each source-water type is only appropriate for the fjord and for the period of study". Are you sure that these chosen SWT are representative for the period of study (late spring, May 2016 and May 2018; early summer, June 2017; early and late fall, September 2014 and November 2017)?

**A seasonality analysis was carried out in order to make sure the SWT definitions are appropriate for the period of study. Insignificant variations were observed in tracers such as $\delta^{18}O$, DIC, TA, DO and $S_P$. The only highly variable tracer was T, which was given the lowest possible weight in the OMP analysis.**

In addition, did you take into account the seasonal variability of the end-members to calculate the OMP and the mixing models?

**As noted above, there is insignificant seasonal variability when it comes to the SWT definitions.**

Line 265: "F= -D $\delta c / \delta x$". Provide the terms of the equation.

**Terms of the equation will be defined in the revised manuscript:**
**F is the diffusion flux in mole $sec^{-1}$ $m^{-2}$**
**D is the diffusion coefficient in $m^2$ $sec^{-1}$**
**C is the concentration of $CO_2$ in mole $m^{-3}$**
**x is the distance in m**

Line 270: The parameterization of Wanninkhof (2014) is recommended for calculations of air-water exchanges in open ocean waters. I think you should include here other parameterization more appropriate for estuarine environments.

**Dinauer and Mucci (2017) analyzed which parameterization was best in the context of the St. Lawrence Estuary system and the parametrization of Wanninkhof (2014) was deemed the most appropriate.**

Line 305: It no is clear to me how you separated these segments for the fjord's surface area. Did you separate by salinity? Distance from the mouth?

**As noted in our response to Reviewer#1's comment, the fjord was divided in segments based on the overall trend of the surface water $pCO_2$ ($pCO_{2(SW)}$) along the fjord (Fig. 4): the first segment includes the larger inner basin (over which $pCO_{2(SW)}$ is much higher than $pCO_{2(air)}$ and decreases rapidly downstream) whereas the second segment encompasses the two outer basins (over which $pCO_{2(SW)}$ is close to $pCO_{2(air)}$ and varies little downstream). Segments will be identified on Figure 1.a.**

Lines 383-385: The discussion of the negative organic alkalinity results are poorly presented. I recommend put more efforts in this subject.

**See above for response to Reviewer#1's comment.**

Lines 414-420: You attributed the average difference between pCO₂(SW-meas) and pCO₂(SW-calc) to the uncertain associated with the carbonic acid dissociation constants. One possible alternative is to calculate the pCO₂(SW-calc) using other available constants to investigate which one fits better with the pCO₂(SW-meas).

**Dinauer and Mucci (2017) investigated which set of carbonic acid dissociation constants returned the most realistic values of pCO₂ in the St. Lawrence Estuary system, which is why the constants from Cai and Wang (1998) were used in this study. As reported in Dinauer and Mucci (2017) other sets of constants return pCO₂(SW-calc) values that differ by as much as ± 300 ppm at salinities below 5.**

Lines 435-446: This paragraph is very interesting, but I missed the comparison with other studies that applied end-member mixing models, contrasting the influences of mixing and biological activities.

**We kindly suggest the referee have a look at the following studies:**

**Dinauer, A., and Mucci, A. (2017). Spatial variability in surface-water pCO₂ and gas exchange in the world's largest semi-enclosed estuarine system: St. Lawrence Estuary (Canada). Biogeosciences, 14(13), 3221-3237.**

**Dinauer, A., and Mucci, A. (2018). Distinguishing between physical and biological controls on the spatial variability of pCO₂: A novel approach using OMP water mass analysis (St. Lawrence, Canada). Marine Chemistry, 204, 107-120.**

Lines 447-457: Where are the results of the fluorometer? I think this section can be strengthened adding with these results. For example you argued that "Additionally, it is interesting to note that NDIC is chronically negative for all sampling months near the 45 km mark." Maybe the fluorescence call tell something.

**We thank the referee for his/her suggestion. We used the CTD profiles from 2014 and 2016 (since the mixing responses are different between these two years) but the fluorescence data do not reveal any significant chronic change that could explain the negative ΔNDIC at the 45 km mark (red line).**

[Figure]

Fig. 1b. Please, provide the title of the Y-right axis. In addition, add the riverine positions in the figure and the estuarine sections you used to calculate the air-water $CO_2$ fluxes.

**The requested information will be added in the revised manuscript.**

Fig. 10. Normally, the comparison of DIC and AOU are performed by calculating the excess of dissolved inorganic carbon (E-DIC), which is difference between the in situ DIC and a theoretical DIC at atmospheric equilibrium. Are there differences comparing ΔNDIC x AOU with E-DIC x AOU?

**It is true that calculating E-DIC is a more conventional way of plotting these data. Nevertheless, there appears to be no notable difference between ΔNDIC x AOU and E-DIC x AOU except for May 2016. This, however, does not change the conclusion of the manuscript given that our focus is on how ΔNDIC changes spatially.**

---

## Author Response (AR2)

**Corrections requested by Anonymous Referee #2**
**(Responses to the referee's comments are in bold)**

- Lines 83 and 543: "Trophic status of the fjord (i.e. whether it is a source or a sink of CO2 to the atmosphere)." As I wrote in the first round of the revision, I am not convinced about this. Trophic status is the degree of nutrient enrichment, and not necessarily related to source and/or sink of CO2.

**The text on both lines was modified to avoid the use of "trophic status".**

- Line 632: I think better to cite the Global Carbon Budget 2019.
Friedlingstein et al 2019. Global Carbon Budget 2019. Earth Syst. Sci. Data, 11, 1783–1838.
https://doi.org/10.5194/essd-11-1783-2019

**We thank the referee for their suggestion. The citation was replaced by the one suggested above.**

**List of all relevant corrections made in the manuscript once accepted**

**TEXT**

- Line 83: Removed "trophic status", sentence now reads "The latter comparison serves to identify the dominant factors, other than physical mixing (i.e., biological activity, gas exchange), that impact the CO2 fluxes at the air-sea interface and modulate their direction and intensity throughout the fjord (i.e. whether it is a source or a sink of CO2 to the atmosphere)."

[revised manuscript text omitted]